# Atomic and electronic modulation of self-supported nickel-vanadium layered double hydroxide to accelerate water splitting kinetics

Dewen Wang[1,2], Qun Li[1,2], Ce Han[1], Qingqing Lu[1,3], Zhicai Xing [1] & Xiurong Yang[1,2]

Herein, ruthenium (Ru) and iridium (Ir) are introduced to tailor the atomic and electronic structure of self-supported nickel-vanadium (NiV) layered double hydroxide to accelerate water splitting kinetics, and the origin of high hydrogen evolution reaction (HER) and oxygen evolution reaction (OER) activities are analyzed at atomic level. X-ray photoelectron spectroscopy and X-ray absorption near-edge structure spectroscopy studies reveal synergistic electronic interactions among Ni, V, and Ru (Ir) cations. Raman spectra and Fourier and wavelet transform analyses of the extended X-ray absorption fine structure indicate modulated local coordination environments around the Ni and V cations, and the existence of V vacancies. The Debye–Waller factor suggests a severely distorted octahedral V environment caused by the incorporation of Ru and Ir. Theoretical calculations further confirm that Ru or Ir doping could optimize the adsorption energy of intermediates in the Volmer and Heyrovsky steps for HER and accelerate the whole kinetic process for OER.

[1] State Key Laboratory of Electroanalytical Chemistry, Changchun Institute of Applied Chemistry, Chinese Academy of Sciences, Changchun 130022, P. R. China. [2] University of Science and Technology of China, Hefei 230026, P. R. China. [3] University of Chinese Academy of Sciences, Beijing 100049, P. R. China. Correspondence and requests for materials should be addressed to Z.X. (email: xingzc@ciac.ac.cn) or to X.Y. (email: xryang@ciac.ac.cn)

Hydrogen, characterized by environmental friendliness and high energy density, has been recognized as an ideal alternative to fossil fuels[1]. With the development of hydrogen-based fuel cells, electrocatalytic water splitting has become a research hotspot for large-scale hydrogen production[2]. The hydrogen evolution reaction (HER) and oxygen evolution reaction (OER) are the two half reactions in the electrocatalytic process, and efficient electrocatalysts are highly desirable to lower the overpotential caused by the sluggish multiple proton-coupled electron transfer in the two reactions[3]. To date, the benchmark electrocatalysts for the HER and OER are Pt- and Ir (or Ru)-based materials, respectively. Considering the low natural abundance and high cost, introducing these noble metals into non-noble metals could not only decrease the dosage of noble metals but also improve the catalytic activity due to the synergistic electronic effect or cause some defect effects[4–6]. Therefore, it is worth undertaking further research to find suitable noble and non-noble materials to improve the catalytic efficiency and expand the application of electrocatalysts.

Layered double hydroxides (LDHs), which can be formulated as $[M^{2+}_{1-x}M^{3+}_x(OH)_2]^{x+}(A^{n-})_{x/n}\cdot mH_2O$, are two-dimensional anionic clays consisting of positively charged brucite-like host layers and exchangeable charge-balancing interlayer anions[7,8]. As ZnCo LDH was first used as an OER electrocatalyst, LDHs based on Fe, Co, Ni, and Mn have been widely studied for OER[9–11]. NiFe-LDHs, as the most active catalysts for OER, have attracted much attention and many methods, such as exfoliation and hybridization, have been applied to further improve their OER activities[12–15]. Recently, the Sun and colleagues[16] prepared monolayer NiV-LDH and it exhibited comparable OER activity to that of NiFe-LDHs. However, both NiV-LDH and the aforementioned LDH electrocatalysts focus on OER, and their HER performance is unsatisfactory due to the large energy barrier of the Volmer step in alkaline conditions[17]. Therefore, there is considerable potential to develop the HER performance of such LDHs and further improve their OER activity. Regarding noble metals, several recent reports have noted that Ru and Ir have suitable adsorption energies for the key reaction intermediates and kinetics for not only OER but also HER[18–22]. Mahmood et al.[21] reported that Ru@C$_2$N has Pt-like activity for HER in alkaline environments. Jiang et al.[22] studied Ir/g-C$_3$N$_4$/nitrogen-doped graphene nanocomposites for HER and OER. These works have inspired us to introduce Ru or Ir into NiV-LDH to design a bifunctional electrocatalyst toward HER and OER. The anionic clay properties of LDH are beneficial to the uniform dispersion of Ru or Ir, thus preventing the negative influence of accumulation on catalytic activity. Furthermore, defects such as vacancies and reconstruction may appear due to Ru or Ir incorporation. The defects can alter the electronic structures and serve as docking sites to capture and stabilize heteroatoms to form heteroatom defect-based motifs with diverse coordination environments and robust stability for efficient and durable HER and OER performance[6,23,24]. In addition, directly growing NiV-LDH on Ni foam and doping Ru and Ir in NiV-LDH can overcome the major drawback (low conductivity) of the LDH materials[25].

We report here that a straightforward one-pot hydrothermal method involving the site-selective incorporation of Ru or Ir into NiV-LDH leads to the excellent bifunctional catalysts NiVRu-LDH and NiVIr-LDH on Ni foam and discuss the results of comparative studies. X-ray photoelectron spectroscopy (XPS) and X-ray absorption near-edge structure (XANES) studies reveal the modified electronic structures of NiVRu-LDH and NiVIr-LDH. Fourier and wavelet transform (FT/WT) analyses of the extended X-ray absorption fine structure (EXAFS) data demonstrate in depth the changed local atomic structure of Ni and V, and the existence of V vacancies. The Debye–Waller factor ($\sigma^2$) suggests a severely distorted octahedral V-Ni/V environment, which subtly modulates local coordination environments in the catalyst. Density functional theory (DFT) studies further reveal that the activation energies of the HER and OER rate-determining steps in alkaline media are optimized through Ru or Ir doping. Both NiVRu-LDH and NiVIr-LDH exhibit improved HER and OER performance compared with that of NiV-LDH in 1 M KOH. Specifically, NiVRu-LDH (Ru in NiVRu-LDH: 1.11 at%) exhibits excellent HER activity with a zero onset overpotential and requires low overpotentials of only 12, 38, and 48 mV to deliver current densities of 10, 100, and 200 mA cm$^{-2}$, respectively, as well as a Tafel slope of 40 mV decade$^{-1}$, which surpasses that of other reported electrocatalysts, including commercial Pt/C. The exchange current density of NiVRu-LDH is 12.6 mA cm$^{-2}$, and the turnover frequency (TOF) is 2.2 s$^{-1}$ at an overpotential of 50 mV. In addition, NiVRu-LDH can maintain 200 mA cm$^{-2}$ for 200 h. NiVIr-LDH (Ir in NiVIr-LDH: 0.62 at%) exhibits the best OER activity: 180 mV overpotential is needed to attain 10 mA cm$^{-2}$ and 200 mA cm$^{-2}$ is sustained for 400 h. A cell voltage of only 1.42 V is required to deliver 10 mA cm$^{-2}$ and maintain high activity for 300 h when NiVRu-LDH and NiVIr-LDH are applied as the cathode and anode in an alkaline electrolyzer, respectively.

## Results

**Synthesis and characterization.** The X-ray diffraction (XRD) patterns of NiV-LDH, NiVRu-LDH, and NiVIr-LDH are shown in Fig. 1a. The diffraction peaks at 11.6, 22.8, 33.5, and 60.5° can be indexed to the characteristic (003), (006), (009), and (110) facets of NiV-LDH[26,27]. With the incorporation of Ru (Ir), no additional diffraction peaks emerge that are associated with the formation of other phases[28]. However, the increased diffraction peak width and reduced peak intensity reveal imperfections in the layers, lattice distortion may be caused by the isomorphic substitution of V by Ru (Ir), and V vacancies appear and recoordinate in the experimental process[29–31].

Moreover, the (003) peak assigned to carbonate-intercalated LDHs, which is also revealed by the Fourier transform infrared (FTIR) spectroscopy analysis (Fig. 1b), corresponds to the stretching vibrations of intercalated carbonates in these samples and is observed at ~680 cm$^{-1}$ and ~1380 cm$^{-1}$. In addition, Raman spectroscopy was performed to obtain information about the chemical identities of these samples (Fig. 1c)[32]. The NiV-LDH samples before and after Ru (Ir) doping exhibit similar spectra, with a strong main peak at ~810 cm$^{-1}$ due to V–O vibration, which is consistent with the FTIR results (the main peak is also located at ~800 cm$^{-1}$)[33,34]. The Raman responses of V–O vibrations for NiVRu-LDH and NiVIr-LDH are clearly weaker than that of NiV-LDH because of the replacement of V by Ru (Ir)[35]. In addition, some of the peaks in inset of Fig. 1c show a certain degree of redshift and some peaks disappear, which can be attributed to two reasons as follows: 1. Size effects[36]. Raman peaks become weaker and blunter and shift slightly to lower wavenumbers as the grain size decreases, so NiVRu-LDH and NiVIr-LDH may have a smaller particle size than that of NiV-LDH. 2. Doping effect and defects[33,35,37]. Defects could be edges, dislocations, cracks, or vacancies in a sample[38]. We suspect that the presence of defects is due to V vacancies, which is consistent with the XRD results and will be further verified later. As depicted in Supplementary Fig. 1, NiVRu-LDH and NiVIr-LDH have much smaller charge transfer resistances than that of NiV-LDH, and the value for NiVRu-LDH is even smaller than that of Pt/C, demonstrating accelerated kinetics after Ru (Ir) doping. In addition, the specific surface areas of NiV-LDH, NiVRu-LDH, and NiVIr-LDH are 36.4, 32.6, and 15.4 m$^2$ g$^{-1}$, respectively (Supplementary Fig. 2). The scanning electron microscopy (SEM)

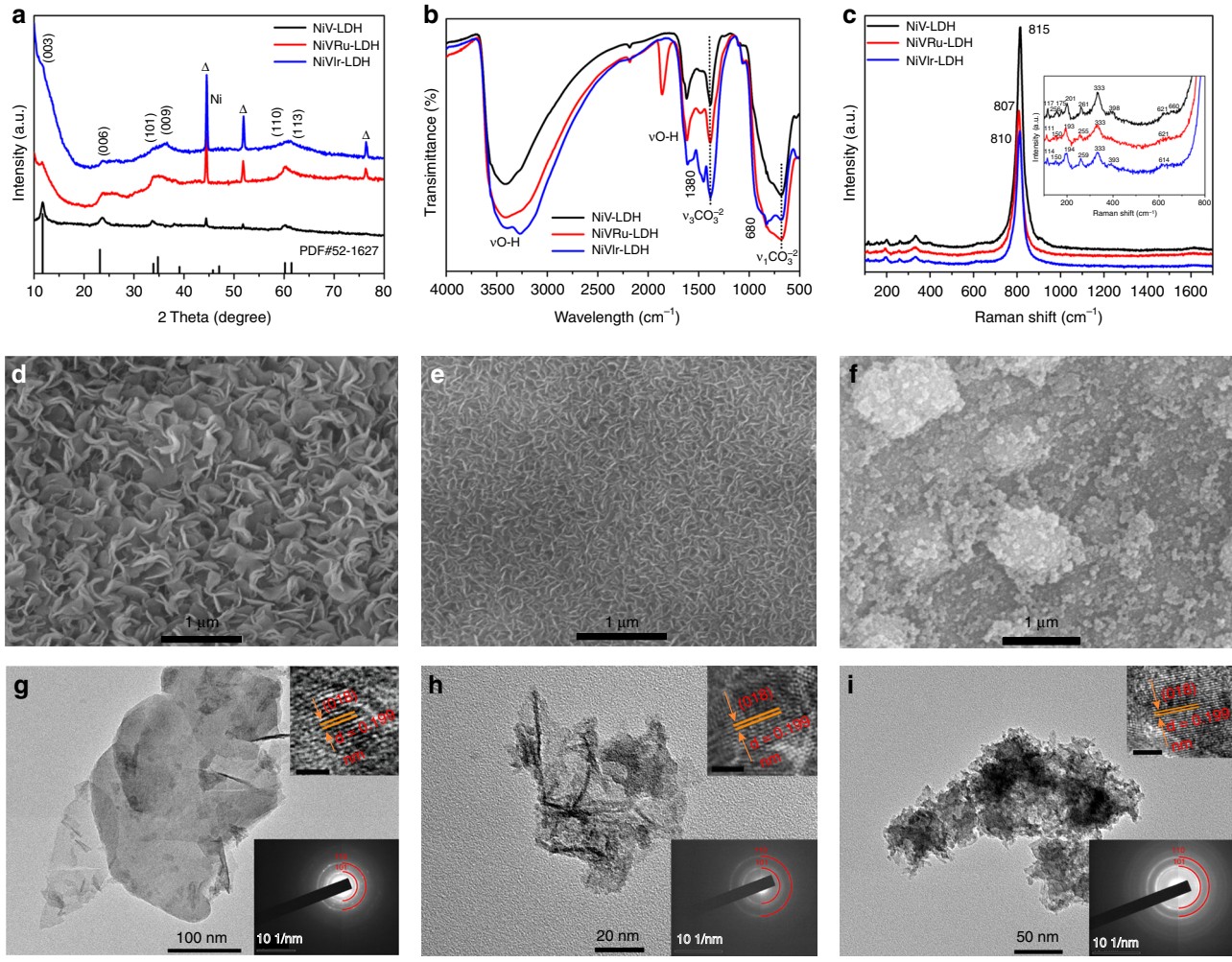

**Fig. 1** XRD patterns, FTIR spectra, Raman spectra, SEM, and TEM images. **a** The XRD patterns of NiV-LDH, NiVRu-LDH, and NiVIr-LDH scraped from Ni foam. **b** FTIR spectra of the electrocatalysts in vacuum condition. **c** Raman spectra of NiV-LDH, NiVRu-LDH, and NiVIr-LDH with 532 nm excitation under ambient air. **d** and **g**, **e** and **h**, **f** and **i** are the SEM and TEM images of NiV-LDH, NiVRu-LDH, and NiVIr-LDH, respectively. The upper right insets of **g**, **h**, and **i** are the HRTEM images of NiV-LDH, NiVRu-LDH, and NiVIr-LDH with scale bars of 1 nm, respectively, and the bottom right insets of **g**, **h**, and **i** are the corresponding SAED patterns

image (Fig. 1d) demonstrates that interconnected NiV-LDH nanosheets grow vertically on the Ni foam, forming a network-like array with obvious ripples and corrugations. The nanosheets become smaller and rougher with the introduction of Ru and Ir (Fig. 1e, f), which is consistent with the Raman results. The corresponding mapping images also indicate that the Ru and Ir distribution increases from sporadic to wide as the content increases (Supplementary Fig. 3–4). The transmission electron microscopy (TEM) images show detailed structural information of the nanosheets. The lattice spacing of 0.199 nm in the high-resolution TEM image can be indexed to the lattice plane of (018) (Fig. 1g). Ru and Ir doping has no influence on the interfringe distance because of the similar ionic radii of $V^{3+}$ (64 pm), $Ir^{3+}$ (68 pm), and $Ru^{3+}$ (68 pm) (Fig. 1h, i). Similar results are also found in the corresponding selected area electron diffraction patterns, which confirm the lack of influence of Ru and Ir doping on the overall crystallinity[31]. In addition, the images for energy-dispersive X-ray (EDX) spectrum elemental analyses indicate the existence of Ni, V, and Ru (Ir) in the nanosheets (Supplementary Figs. 5–6). The spatial distribution of every element in NiVRu-LDH and NiVIr-LDH is further identified by electron energy loss spectroscopy (EELS) mapping and Fig. 2a, b indicate the homogenous Ru and Ir doping in the nanosheets.

**Understanding atomic modification and electronic interaction.** XPS was used to analyze the surface compositions of the electrocatalysts. The two peaks of Ni $2p_{3/2}$ and Ni $2p_{1/2}$, located at 855.8 and 873.5 eV, respectively, are the characteristic features of $Ni^{2+}$ (Fig. 2c), accompanied by two shakeup satellite peaks at 861.9 and 880.1 eV[39,40]. V $2p$ can be fitted into V $2p_{3/2}$ and V $2p_{1/2}$ (Fig. 2d), and the three peaks of V $2p_{3/2}$ located at 516.0, 517.1, and 518.1 eV correspond to $V^{3+}$, $V^{4+}$, and $V^{5+}$, respectively, suggesting that $V^{3+}$ has been partially oxidized to $V^{4+}$ and $V^{5+}$ during the synthesis process[16,41,42]. It is worth noting that the binding energies of both Ni and V in NiVRu-LDH and NiVIr-LDH have slight negative shifts of 0.5 eV compared with that of NiV-LDH. The results indicate the synergistic electronic interactions among Ni, V, and Ru (Ir) cations, so the introduced Ru (Ir) has a strong influence on the electronic structure of NiVRu-LDH and NiVIr-LDH. The peaks at 463.9 and 485.5 eV can be attributed to Ru $3p_{3/2}$ and Ru $3p_{1/2}$ of $Ru^{3+}$ (Fig. 2e), respectively, confirming the presence of Ru in NiVRu-LDH with a valence state of 3+[43]. The Ir $4f$ region consists of $4f_{7/2}$ and $4f_{5/2}$ (Fig. 2f). The peaks at 61.8 and 66.8 eV can indexed to Ir $4f_{7/2}$ and Ir $4f_{5/2}$ of $Ir^{4+}$, respectively, and the peaks at 62.4 and 65.6 eV are Ir $4f_{7/2}$ and Ir $4f_{5/2}$ of $Ir^{3+}$; the $Ir^{3+}$-dominated structure also indicates partial oxidation[44,45].

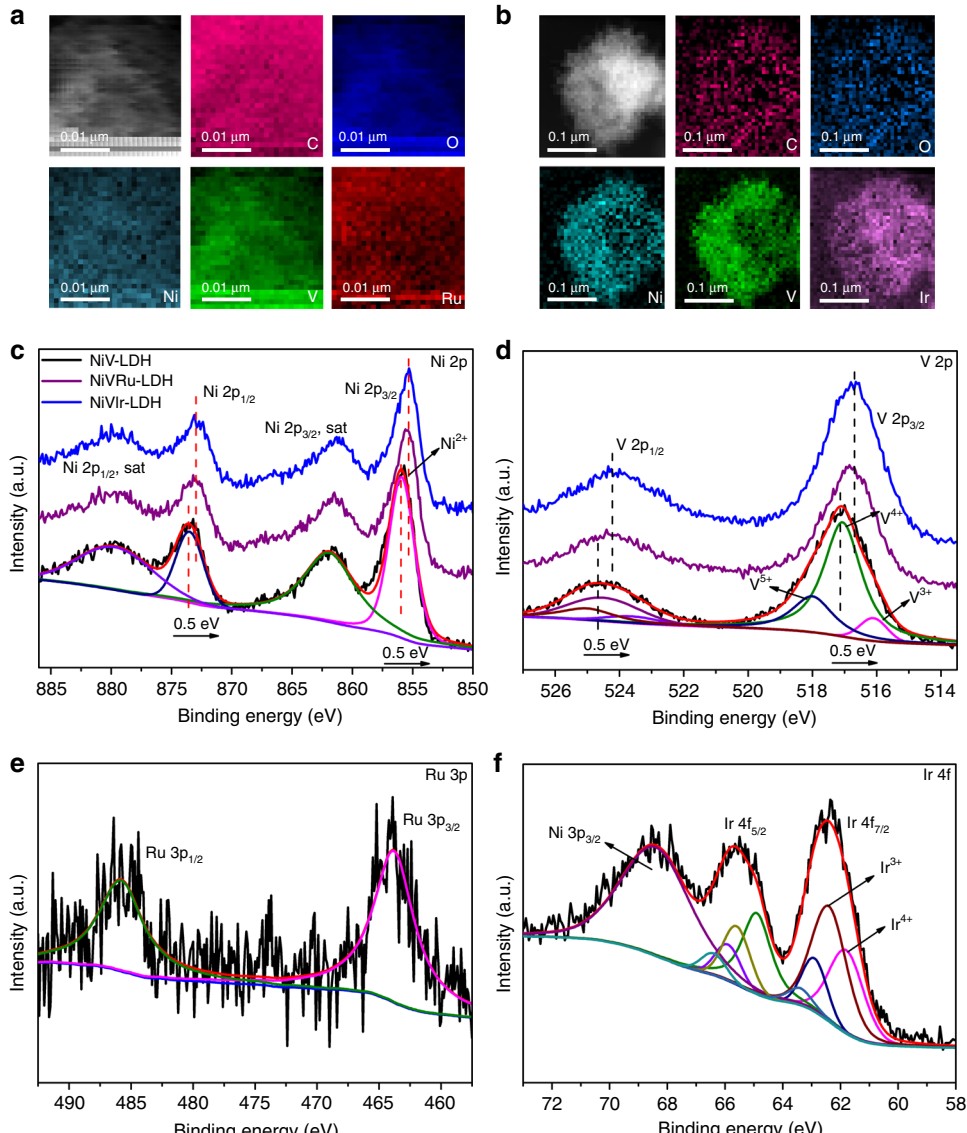

**Fig. 2** HAADF-STEM images and XPS spectra. HAADF-STEM images of **a** NiVRu-LDH and **b** NiVIr-LDH, and the corresponding EELS mappings images. The high-resolution XPS signals of NiV-LDH, NiVRu-LDH, and NiVIr-LDH for **c** Ni 2p and **d** V 2p. The high-resolution XPS signals of Ru 3p in **e** NiVRu-LDH and **f** Ir 4 f in NiVIr-LDH

In addition, the peak of Ni $3p_{3/2}$ overlaps within the Ir $4f$ spectral window[46].

To clarify the V vacancies and the incorporation effect of Ru (Ir) on the local atomic coordination and electronic structure of NiV-LDH, XANES spectroscopy was conducted. As shown in Fig. 3a, b, the Ni K-edge spectra of the three samples are quite similar; however, there are significant differences in the V K-edge spectra, suggesting that the doping of Ru (Ir) has a stronger effect on V than that on Ni. Specifically, the V K-edge XANES spectra of the sample exhibit intense pre-edge peaks (inset in Fig. 3b), indicating the distorted coordination environment around V atoms in these materials[47,48]. More interestingly, NiVRu-LDH and NiVIr-LDH show a higher pre-edge peak than that of NiV-LDH in the V K-edge XANES, implying that Ru (Ir) incorporation brings a higher degree of octahedral geometry distortion at the V sites in NiVRu-LDH and NiVIr-LDH compared with those in NiV-LDH. Similar results are found in the WT analyses of the Ni K-edge and V K-edge data (Fig. 3e, f). There are no obvious differences, except for the weaker peak intensity at ~2.5–3.0 Å of NiVRu-LDH and NiVIr-LDH than

that of NiV-LDH in the EXAFS WT map, indicating the existence of Ni or V vacancies. In contrast, the peak intensities at ~3.0 Å of NiVRu-LDH and NiVIr-LDH are significantly less than that of NiV-LDH, which also shows that the change in V is larger than that in Ni. The different oscillation amplitudes in the corresponding Ni K-edge and V K-edge $k^3\chi(k)$ oscillation curves (Supplementary Fig. 7) reveal a structural change in the coordination environment of Ni and V atoms[49]. Thus, the Ni K-edge and V K-edge R-space spectra (Fig. 3c, d and Supplementary Table 1) provide detailed information about the coordination number (C.N.)[50]. The FT curves of the Ni K-edge data exhibit two prominent coordination peaks at 1.5 and 2.7 Å, which are attributed to the Ni-O peak and Ni-Ni/V peak, and the C.N. values of Ni-Ni/V in NiVRu-LDH (4.6) and NiVIr-LDH (4.2) are slightly reduced compared with that in NiV-LDH (5.1). Similarly, the FT curves of the V K-edge data display prominent V–O peak at 1.3 Å in these three samples, and V-Ni/V peak at 2.97 Å in NiV-LDH, 2.93 Å in NiVRu-LDH, and 2.84 Å in NiVIr-LDH (Fig. 3d). The C.N. values of V-Ni/V in NiVRu-LDH (3.8) and NiVIr-LDH (2.8) are obviously reduced compared with

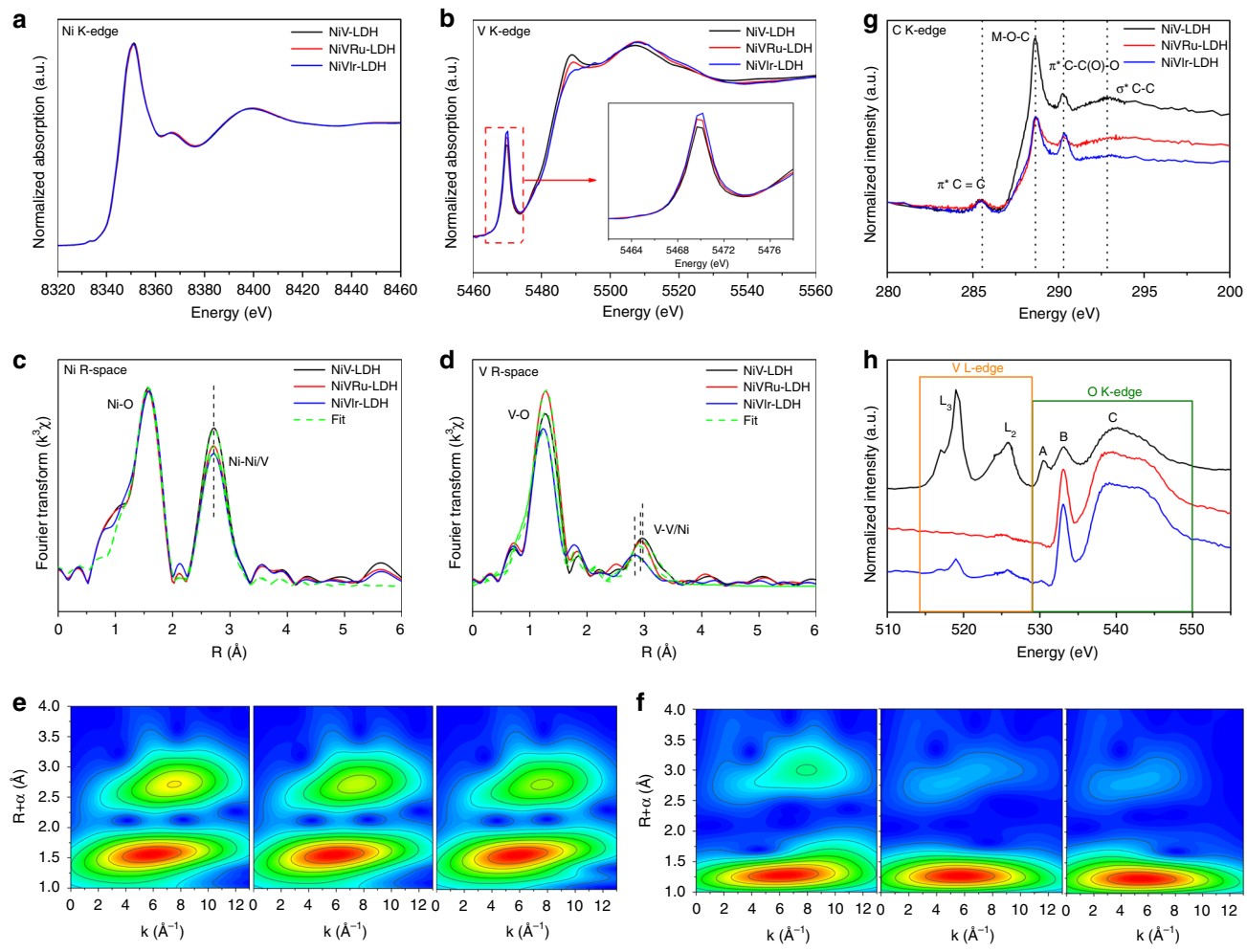

**Fig. 3** XANES and WT-EXAFS spectra. **a** Ni and **b** V K-edge XANES spectra. **c** Ni and **d** V K-edge extended XANES oscillation functions $k^3\chi$ ($k$). WT-EXAFS of **e** Ni and **f** V of the as-prepared NiV-LDH, NiVRu-LDH, and NiVIr-LDH (from left to right). **g** C K-edge XANES spectra. **h** V L-edge and O K-edge XANES spectra

that in NiV-LDH (5.3); a larger $\sigma^2$ for the V-Ni/V of NiVRu-LDH (0.0172) and NiVIr-LDH (0.0186) was obtained compared with the corresponding value for NiV-LDH (0.0123), suggesting a severely distorted octahedral V-Ni/V environment[51]. Together, these results confirm that V vacancies do exist in the material. The C K-edge XANES spectra in Fig. 3g show two peaks at ~285.4 and 292.7 eV, which can be assigned to $\pi^*$C=C and $\sigma^*$C–C, respectively[52]. Notably, NiVRu-LDH and NiVIr-LDH show a distinct decrease in peak intensity at ~288.6 eV (which is assigned to M–O–C bonds; M = Ni/V) compared with that of NiV-LDH[53]. On the basis of both the XANES and EXAFS data for Ni and V above, the M here should be V. This clearly indicates the absence of V in NiVRu-LDH and NiVIr-LDH. Based on previous reports, another peak at ~290.2 eV is assigned to carbonate, which originates from the interlayer carbonate in NiV-LDH[12]. This observation is consistent with the FTIR results. The positions of the V L- and O K-edges make their spectra partially overlapped, so the combined spectra of the V L- and O K-edges are presented in Fig. 3h[54]. Three peaks (A, B, and C) are observed in the O K-edge XANES spectra[55]. Peak A at 530.5 eV and the broad peak C at 539 eV are assigned to $\pi^*$C=O and $\sigma^*$C–O, respectively. Peak B at near 533.1 eV may be assigned to $\pi^*$C–O. In the V L-edge spectra, the two broad peaks centered at 519 and 525.9 eV are the $L_3$ and $L_2$ peaks, which are assigned to V $2p_{3/2}$ and V $2p_{1/2}$ transitions, respectively, and which are consistent with the XPS results[54,56]. In sharp contrast, the peaks of $L_2$ and $L_3$ of V are noticeable in NiV-LDH but are almost

absent in NiVRu-LDH and NiVIr-LDH, which strongly proves that vacancies of V in NiVRu-LDH and NiVIr-LDH exist.

**Electrocatalytic activity toward HER.** The HER performances of the as-obtained NiV-LDH, NiVRu-LDH, NiVIr-LDH, and Pt/C were evaluated in 1 M KOH at a scan rate of 5 mV s$^{-1}$ by a typical three-electrode system with $iR$ correction and the potentials were calibrated vs. the reversible hydrogen electrode (RHE). As shown in Fig. 4a, b, NiV-LDH exhibits low HER activity and 209 mV overpotential is needed to deliver 10 mA cm$^{-2}$. In sharp contrast, the activities of NiVRu-LDH and NiVIr-LDH are substantially improved compared with that of pure NiV-LDH. To attain current densities of 10, 100, and 200 mA cm$^{-2}$, NiVIr-LDH requires overpotentials of 47, 159, and 213 mV, respectively. NiVRu-LDH exhibits excellent HER performance; only 12, 38, and 48 mV is needed to obtain a current density of 10, 100, and 200 mA cm$^{-2}$, respectively, indicating even better performance than that of the Pt/C electrocatalyst (12, 81, and 149 mV for 10, 100, and 200 mA cm$^{-2}$, respectively) and previously reported HER electrocatalysts (Supplementary Table 2). The HER performance of NiVRu-LDH with a Ru content of 1.11 at% is better than that of any NiVRu-LDH samples with other Ru contents, as shown in Supplementary Fig. 8.

The Tafel plots of the corresponding polarization curves are shown in Fig. 4c, which can indicate the electrochemical reaction kinetics of the electrocatalysts[57]. The Tafel slope of NiVRu-LDH

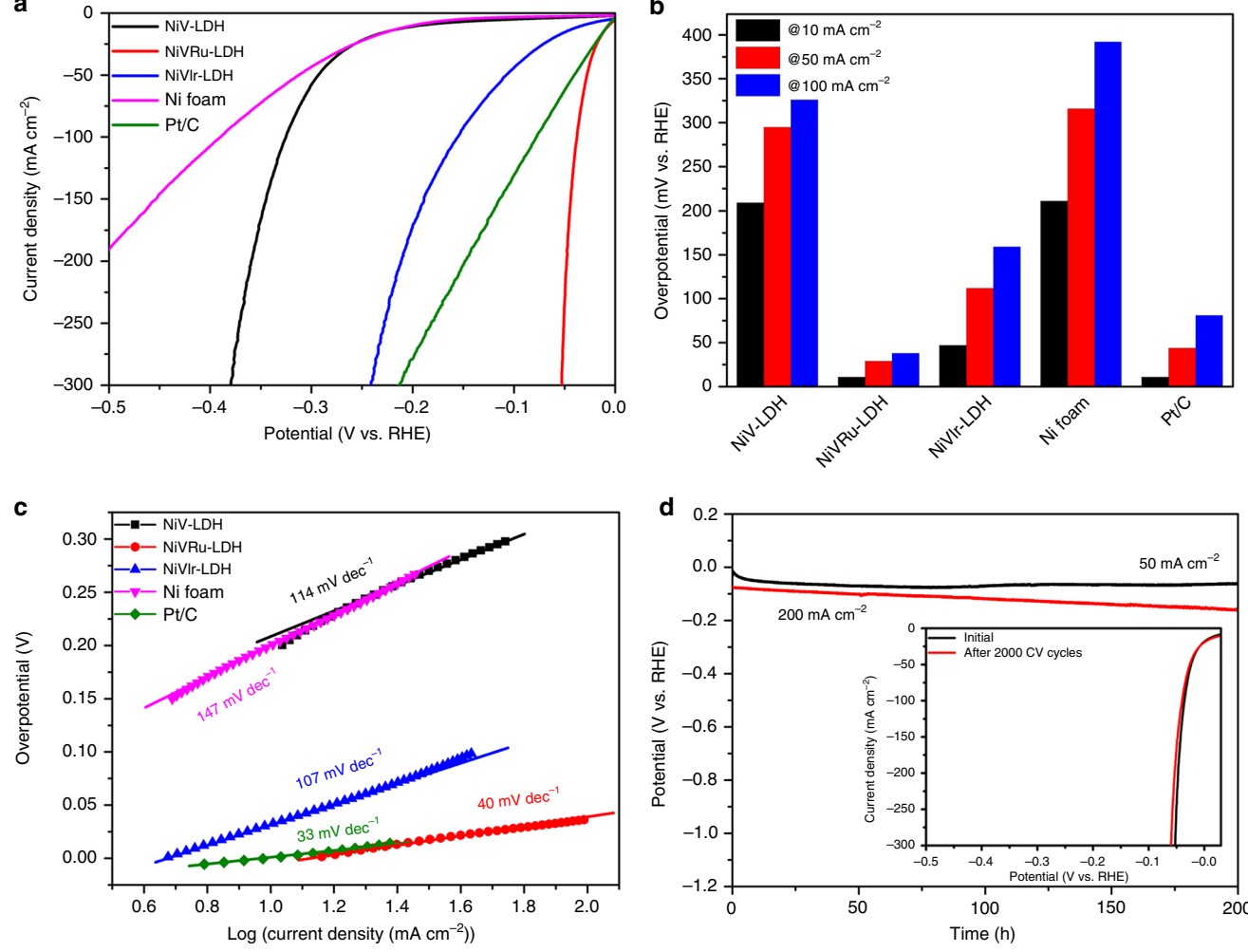

**Fig. 4** HER polarization curves, Tafel plots, and long-term stability. **a** HER polarization curves, **b** the overpotentials at 10, 50, and 100 mA cm$^{-2}$, and **c** the corresponding Tafel plots of the as-obtained NiV-LDH, NiVRu-LDH, NiVIr-LDH, Ni foam, and Pt/C. **d** The chronopotentiometric curves of NiVRu-LDH for the HER test at 50 and 200 mA cm$^{-2}$. Inset: polarization curves of NiVRu-LDH for the durability test after 2000 CV cycles

is 40 mV decade$^{-1}$, which is much smaller than those of NiV-LDH (114 mV decade$^{-1}$), NiVIr-LDH (107 mV decade$^{-1}$), and Ni foam (147 mV decade$^{-1}$). This result shows that the hydrogen evolution rate will rapidly increase with the application of overpotential and the reaction process follows the Volmer–Heyrovsky mechanism[58,59]. The Tafel slope for Pt/C is 33 mV decade$^{-1}$, which belongs to the Volmer–Tafel process. The exchange current density of NiVRu-LDH is estimated to be 12.6 mA cm$^{-2}$ (Supplementary Fig. 9), which is much better than that of other materials, including Pt/C (9.33 mA cm$^{-2}$). Furthermore, TOF was used to measure the intrinsic catalytic activity of each active site (Supplementary Fig. 10–11). The TOF of NiVRu-LDH (2.2 s$^{-1}$) at an overpotential of 50 mV is ~30 times higher than that of NiV-LDH (0.074 s$^{-1}$), which is also higher than those of NiVIr-LDH (0.29 s$^{-1}$) and Pt/C (0.96 s$^{-1}$). Then, the electrochemical double-layer capacitance ($C_{dl}$) was determined by performing cyclic voltammetry (CV) at various scan rates to evaluate the active sites of the electrocatalyst by calculating the electrochemically active surface area. As shown in Supplementary Fig. 12, NiVRu-LDH has the largest $C_{dl}$ of 35.2 mF cm$^{-2}$, which is better than those of NiVIr-LDH (19 mF cm$^{-2}$) and NiV-LDH (1.7 mF cm$^{-2}$).

The electrocatalytic stability is also an important figure of merit in judging an excellent electrocatalyst. The durability of NiVRu-LDH was first tested by continuous CV scans at a scan rate of

50 mV s$^{-1}$ in 1 M KOH. As depicted in Fig. 4d, the polarization curve of NiVRu-LDH exhibits negligible differences compared with the initial curve after 2000 CV scans. Next, a long-term electrocatalytic HER process was carried out at current densities of 50 and 200 mA cm$^{-2}$, and the steady HER overpotentials were retained for 200 h. After a durability test at 50 mA cm$^{-2}$ for 200 h, the original morphology was essentially retained (Supplementary Fig. 13), highlighting the superior structural robustness during the electrocatalytic HER process.

**Electrocatalytic activity toward OER and overall water splitting.** Then, the influences of Ru and Ir doping on the OER performance of NiV-LDH were investigated in 1 M KOH and RuO$_2$ deposited on Ni foam was used as a reference. To minimize the effect of the capacitive current originating from Ni ion oxidation, the polarization curves were collected from high to low potentials with a scan rate of 1 mV/s. As shown in Fig. 5a, b, the OER activities of NiV-LDH, NiVRu-LDH, and NiVIr-LDH are all better than that of RuO$_2$. Furthermore, the activities of both NiVRu-LDH and NiVIr-LDH are obviously improved compared with that of NiV-LDH. To attain a current density of 10 mA cm$^{-2}$, NiV-LDH needs an overpotential of 200 mV, whereas the NiVRu-LDH needs 190 mV. Remarkably, NiVIr-LDH with an Ir content of 0.62 at% exhibits the best OER performance, requiring only 180 mV to obtain a current density

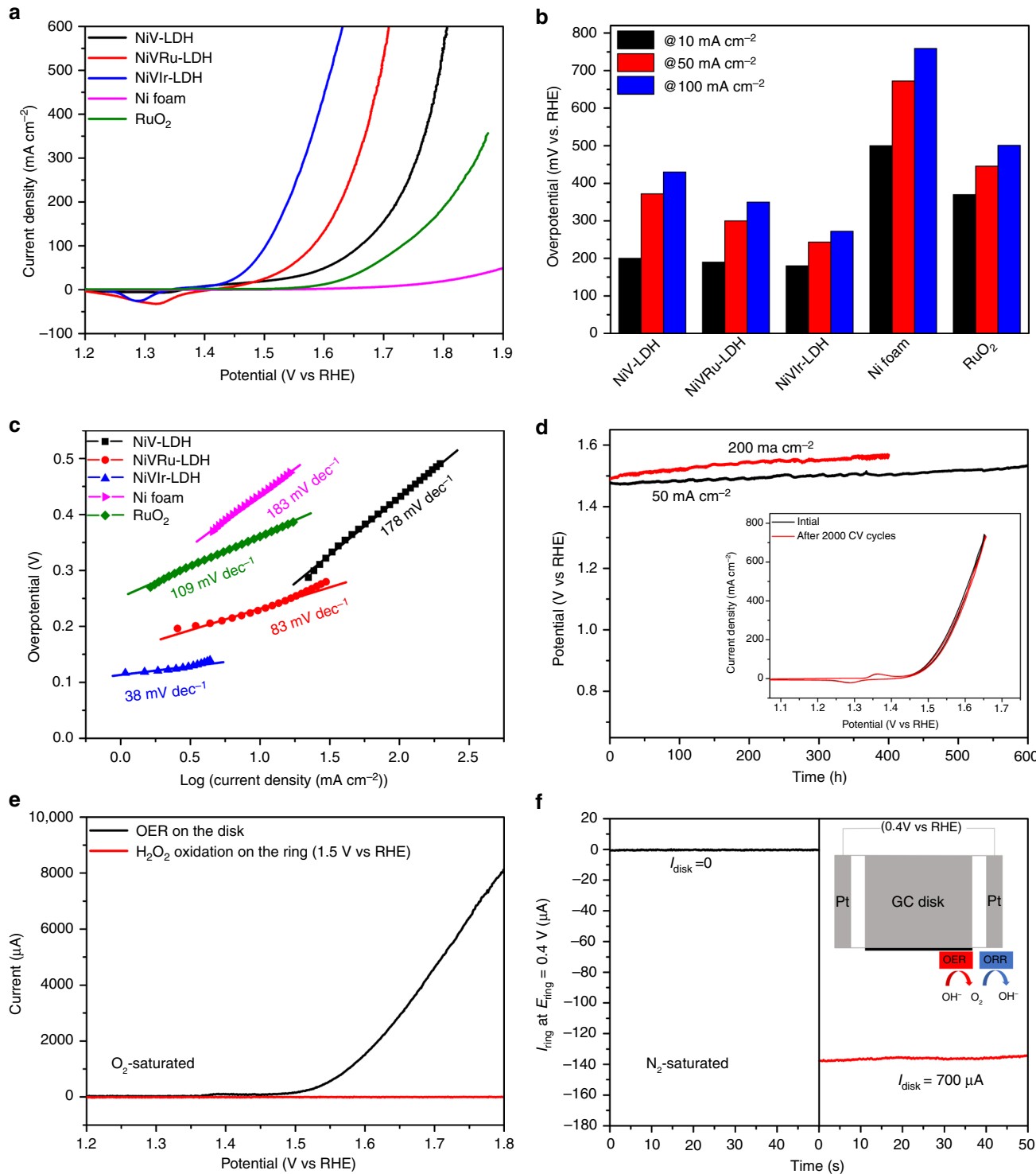

**Fig. 5** OER polarization curves, Tafel plots, and long-term stability. **a** OER polarization curves, **b** the overpotentials at 10, 50, and 100 mA cm$^{-2}$, and **c** the corresponding Tafel plots of the as-obtained NiV-LDH, NiVRu-LDH, NiVIr-LDH, Ni foam, and RuO$_2$. **d** The chronopotentiometric curves of NiVIr-LDH for the OER test at 50 and 200 mA cm$^{-2}$. Inset: polarization curves of NiVIr-LDH for the durability test after 2000 CV cycles. **e** Detection of H$_2$O$_2$ generated using RRDE measurement. The ring current of NiVIr-LDH on a RRDE (1500 r.p.m.) in O$_2$-saturated 1 M KOH solution (ring potential: 1.50 V). **f** The ring current of NiVIr-LDH on a RRDE (1500 r.p.m.) in N$_2$-saturated 1 M KOH solution (ring potential: 0.40 V)

of 10 mA cm$^{-2}$ (Supplementary Fig. 14) and 272 mV to obtain 100 mA cm$^{-2}$, demonstrating a substantial improvement in OER activity compared with that of NiV-LDH (100 mA cm$^{-2}$ at 430 mV). The activity is also better than those of other OER electrocatalysts in 1 M KOH (Supplementary Table 3). A very

small Tafel slope of 38 mV dec$^{-1}$ was measured for NiVIr-LDH, which shows the fastest kinetics among these samples (Fig. 5c). NiVIr-LDH has strong stability during the OER process; the polarization curve shows a negligible change after 2000 CV scans and the material displays long-term electrochemical durability for

600 h at 50 mA cm$^{-2}$ and 400 h at 200 mA cm$^{-2}$, respectively (Fig. 5d). The original morphology of NiVIr-LDH remained intact after testing for 600 h (Supplementary Fig. 15).

To obtain more detailed information about the reaction mechanism, a rotating ring-disk electrode (RRDE) with a Pt ring electrode potential of 1.50 V vs. RHE was used to detect whether peroxide intermediates formed during the OER process[60,61]. As shown in Fig. 5e, no detectable current from peroxide oxidation at the Pt ring is observed, suggesting the negligible formation of peroxide intermediates and therefore a desirable four-electron pathway for water oxidation. An RRDE can also be used to quickly screen the approximate Faradaic efficiency (FE) of O$_2$ production ($\varepsilon$)[62]. We employed the RRDE in N$_2^-$ saturated 1 M KOH with a ring potential of 0.40 V to reduce molecular O$_2$, rendering a continuous OER (disk electrode) → ORR (oxygen reduction reaction, ring electrode) process (Fig. 5f)[63,64]. With a disk current of 700 µA, O$_2$ molecules generated from the NiVIr-LDH surface on the disk electrode sweep across the surrounding Pt ring electrode held at an ORR potential and are rapidly reduced. Consequently, a ring current of ~140 mA (700 mA × 0.2; RRDE collecting efficiency $N = 0.2$) was detected, suggesting that the oxidation current can be fully attributed to the OER with a high $\varepsilon$ of >99%.

Considering that the self-supported NiVRu-LDH and NiVIr-LDH materials have such excellent HER and OER properties, respectively, we employed NiVRu-LDH as the cathode and NiVIr-LDH as the anode to fabricate an electrolyzer (NiVIr-LDH‖NiVRu-LDH) toward overall water splitting. The NiVIr-LDH‖NiVRu-LDH couple exhibits excellent performance: only a 1.42 V cell voltage is needed to deliver a 10 mA cm$^{-2}$ current density (Fig. 6a); this value is much lower than that of RuO$_2$‖Pt/C (10 mA cm$^{-2}$ at 1.50 V) and most reported articles (Fig. 6b, Supplementary Table 4), and even comparable to the performance of two well-known materials, namely the NiFe LDH-NS@DG catalyst reported by Yao and colleagues[23] (20 mA cm$^{-2}$ at 1.5 V; our catalyst obtained 20 mA cm$^{-2}$ at 1.49 V) and the FeP/Ni$_2$P catalyst reported by Ren and colleagues[65] (10 mA cm$^{-2}$ at 1.42 V). In addition, the electrolyzer demonstrates good long-term catalytic durability, sustaining constant galvanostatic electrolysis for up to 300 h at 10 mA cm$^{-2}$ with negligible degradation (Fig. 6c), suggesting excellent potential for practical applications. Excitingly, the electrolyzer could be driven by a single-cell AAA battery with a nominal voltage of ~1.5 V at room temperature, as shown in the inset of Fig. 6a and Supplementary Movie 1. The FE of NiVRu-LDH and NiVIr-LDH for the HER and OER was measured quantitatively from the total amount of charge passed through the cell during electrolysis and the total amount of evolved gas recorded by the pressure sensor[66]. The amount of experimentally generated H$_2$ and O$_2$ matches well with the theoretically calculated amount under the total charge during the electrolysis process (Fig. 6d), suggesting that the FE is close to 100% for the HER and OER, with the ratio of H$_2$ and O$_2$ being close to 2:1.

**First-principles calculations**. To investigate the original relationship between the excellent activity and the atomic and electronic structure, DFT methods were performed to obtain the free energy diagrams of every step in HER and OER (Supplementary Figs. 16–19 and Supplementary Table 5). The Volmer–Heyrovsky mechanism of NiVRu-LDH for HER is displayed in Fig. 7a[58]. It can be seen from Fig. 7b that NiV-LDH has a large water dissociation energy barrier ($\Delta G$(H$_2$O)) of −0.5 eV in the prior Volmer step and $\Delta G$(H$_2$O) of NiVIr-LDH and NiVRu-LDH is −0.27 and 0.05 eV, respectively. In addition, the hydrogen adsorption free energy ($\Delta G$(H)) of NiVRu-LDH is −0.25 eV,

which is also better than those of NiV-LDH (−0.55 eV) and NiVIr-LDH (−0.46 eV). The results suggest that Ru and Ir doping can accelerate both Volmer and Heyrovsky steps in alkaline media, and that Ru is more conducive to HER[21]. The proposed OER process of NiVIr-LDH in alkaline conditions consists of four elementary stages: M*, M-OH, M-O, and M-OOH (Fig. 7c)[67]. The calculated free energy of each elementary step is shown in Fig. 7d. Every step of NiVRu-LDH and NiVIr-LDH is accelerated compared with those of NiV-LDH. Specifically, Ru doping is beneficial to the M-OOH step and Ir doping is more conducive to the M-OH and M-O process; ultimately, NiVIr-LDH has the best OER performance. The above theoretical calculations are consistent with the experimental results, which fully prove that the modulated atomic and electronic structure of NiV-LDH by Ru and Ir doping can achieve higher HER and OER activity.

## Discussion

In conclusion, we synthesized self-supported NiV-LDH, NiVRu-LDH, and NiVIr-LDH on Ni foam, and comparative studies demonstrate that synergistically modulating the atomic and electronic structure of NiV-LDH by doping Ru or Ir with optimal doping levels could boost both HER and OER activity in alkaline solutions. NiVRu-LDH and NiVIr-LDH feature an apparently smaller nanosheet size and charge transfer resistance than those of NiV-LDH. XRD patterns and Raman spectra reveal possible lattice distortion caused by an isomorphic substitution of V by Ru (Ir), and V vacancies appear, which is confirmed by the FT and WT analyses of EXAFS data, and the $\sigma^2$ results suggest a severely distorted octahedral V environment caused by Ru or Ir doping. The results of XPS and XANES reveal the synergistic interaction among Ni, V, and Ru (Ir) cations, derived from quite different valence electronic configurations. The as-obtained self-supported NiVRu-LDH and NiVIr-LDH have better exchange current densities and TOF values than those of NiV-LDH and exhibit excellent HER and OER activities and stabilities. The NiVIr-LDH‖NiVRu-LDH electrolyzer only needs a 1.42 V cell voltage to deliver a 10 mA cm$^{-2}$ current density. The experimental results and the DFT calculations verify that Ru and Ir doping can reduce the energy barrier of both the Volmer and Heyrovsky steps in HER and accelerate every step of OER, thus promoting the activities of the HER and OER. This work provides an in-depth analyses of the changes in the atomic and electronic structures of NiV induced by the doping of Ru and Ir, which can provide valuable information for related work.

## Methods

**Materials**. Ni foam was purchased from Shenzhen Green and Creative Environmental Science and Technology Co. Ltd. KOH and absolute alcohol were purchased from Beijing Chemical Corp. Ni(NO$_3$)$_2$·6H$_2$O was purchased from Fuchen Chemical Reagents Factory. VCl$_3$ was purchased from the Energy Chemical. Urea and IrCl$_3$·H$_2$O were purchased from Aladdin. Pt/C (20 wt% Pt on Vulcan XC-72R), RuCl$_3$, and Nafion (5 wt%) were purchased from Sigma-Aldrich. All chemicals were used as received without further purification. The water used throughout all experiments was purified through a Millipore system.

**Synthesis of NiV-LDH, NiVRu-LDH, and NiVIr-LDH**. NiV-LDH was synthesized by a simple one-step hydrothermal method. Briefly, a piece of Ni foam (2 cm × 3 cm) was ultrasonically cleaned in HCl solution for 10 min to dissolve the surface oxide layer and then washed with deionized water for several times. Then, 0.6 mmol of Ni(NO$_3$)$_2$·6H$_2$O, 0.48 mmol of VCl$_3$, and 4 mmol of urea (the role of urea can be seen in Supplementary Note 1) were dispersed in 30 mL deionized water and stirred for 30 min. The cleaned Ni foam was submerged into the solution and transferred into a 50 mL Teflon-lined stainless steel autoclave. The autoclave was heated to 120 °C for 12 h in an electric oven and then cooled to room temperature. The product was washed with deionized water for several times and then dried in an oven at 60 °C to obtain NiV-LDH. The synthesis processes of NiVRu-LDH and NiVIr-LDH were the same as that of NiV-LDH, except for the addition of RuCl$_3$ and IrCl$_3$·xH$_2$O, respectively. To synthesize NiVRu-LDH and NiVIr-LDH with

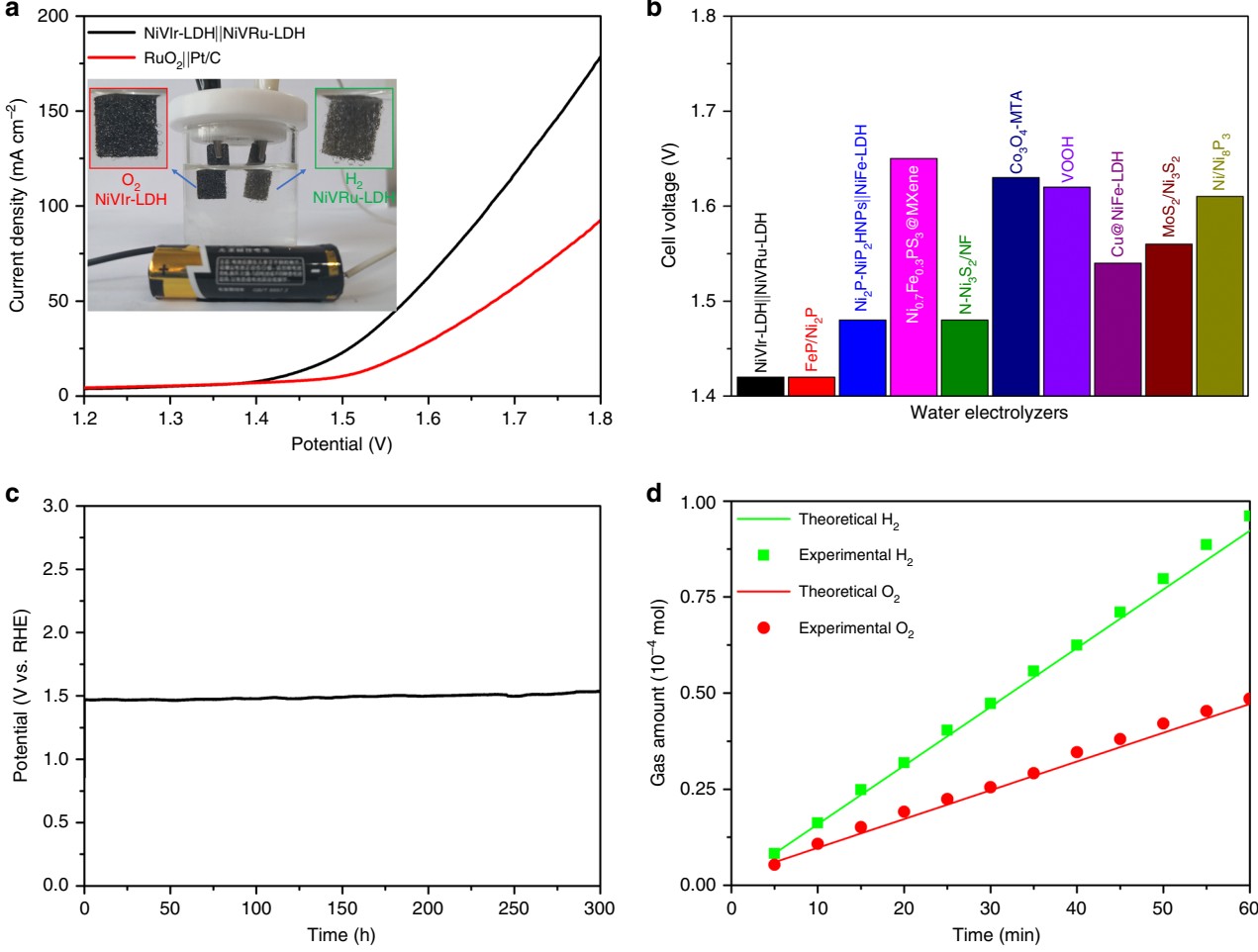

**Fig. 6** Polarization curves, long-term stability, and Faradaic efficiency. **a** The polarization curves of NiVIr-LDH||NiVRu-LDH and RuO₂||Pt/C toward overall water splitting. The inset is the electrolyzer driven by a single-cell AAA battery. **b** Comparison of the required voltage at a current density of 10 mA cm⁻² for the NiVIr-LDH||NiVRu-LDH with other state-of-the-art bifunctional catalysts. **c** The chronopotentiometric curve of NiVIr-LDH||NiVRu-LDH for overall water splitting at a constant current density of 10 mA cm⁻². **d** The amount of gas theoretically calculated and experimentally measured vs. time for HER and OER of NiVRu-LDH and NiVIr-LDH, respectively

different Ru and Ir contents, the dosages of RuCl₃ (IrCl₃·xH₂O) were 0.06, 0.12, 0.18, 0.24, 0.3, and 0.36 mmol for samples 1~ 6, respectively. The mass loadings of NiV-LDH, NiVRu-LDH, and NiVIr-LDH were 2.6, 1.2, and 0.7 mg cm⁻², respectively. The contents of Ru and Ir were determined by XPS. NiV-LDH, NiVRu-LDH with 1.11 at% Ru, and NiVIr-LDH with 0.62 at% Ir were used for material characterizations.

**Preparation of Pt/C and RuO₂ electrodes**. A total of 5 mg of Pt/C (RuO₂) was ultrasonically dispersed in 0.5 mL of deionized water, 0.495 mL of ethanol, and 0.05 mL of 5 wt% Nafion® solution, and the ink was transferred onto a Ni foam electrode via a controlled drop casting method. The loading amounts of Pt/C and RuO₂ were both 1.2 mg cm⁻².

**Material characterizations**. Powder XRD data were acquired with a RigakuD/MAX 2550 diffractometer with Cu Kα radiation (λ = 1.5413). Raman spectra were collected on a Smart System HR Evolution (Horiba JobinYvon, France) using 532 nm laser source. FTIR spectra were acquired by a Bruker VERTEX70 spectrometer in the range of 4000−5000 cm⁻¹ (Ettlingen, Germany). SEM and the corresponding elemental mapping and EDX analysis measurements were performed with an XL30 ESEM FEG microscope at an accelerating voltage of 20 kV. TEM measurements were made with a Hitachi H-8100 electron microscope (Hitachi, Tokyo, Japan) with an accelerating voltage of 200 kV. The scanning TEM imaging and EELS mapping were carried out on a cubed FEI Titan G² electron microscope equipped with both a probe corrector and a monochromator operated at 200 kV. The probe convergence angle is 21.4 mrad, probe size 1 Å, and a best energy resolution of 0.4 eV, as measured from the full-width-at-half-maximum of the zero-loss peak spectrometer entrance aperture: 2.5 mm channel dispersion: 0.5 eV/Chanel. XPS measurements were performed with an ESCALAB MK II X-ray photoelectron spectrometer by using Mg as the excitation source. The

Brunauer–Emmett–Teller surface area and Barrett–Joyner–Halenda pore size distribution were measured on a Quantachrome NOVA 1000 system at liquid N₂ temperature. X-ray absorption fine structure spectroscopy experiment was carried out at 1W2B end station, Beijing Synchrotron Radiation Facility.

**Electrochemical measurements**. All electrochemical measurements were performed with a CHI660e electrochemical analyzer (CH Instruments, Inc., Shanghai) at room temperature. NiV-LDH, NiVRu-LDH, and NiVIr-LDH were directly used as the working electrode, a saturated calomel electrode (SCE) was used as the reference electrode, and a graphite rod was used as the counter electrode. In this study, E(RHE) = E(SCE) + 0.242 V + 0.059 pH. The ohmic potential drop losses that arise from the solution resistance were corrected by *iR* compensation. Tafel plots of the overpotential vs. log (*j*) were recorded, and the linear portions at low overpotential were fitted to the Tafel equation (η = a + b log *j*, where η is the overpotential, *j* is the cathodic current density, and *b* is the Tafel slope). The stability tests of NiV-LDH, NiVRu-LDH, and NiVIr-LDH were also performed using a typical three-electrode system. The double-layer capacitance (*C*_dl) values at the solid–liquid interface of materials were measured by CV between 0.15 V and 0.35 V vs. RHE in 1.0 M KOH, where the current response should be only due to the charging of the double layer. RRDE voltammetry experiments were performed using a CHI760E Electrochemical Analyzer (CH Instruments, China), a speed control unit (Princeton Applied Research Model 636 Electrode Rotator) (the rotating speed of the RRDE was held at 1600 r.p.m.) and a PINE RRDE with a glassy carbon (GC) disk and Pt ring. The three-electrode cell consisted of a Ag/AgCl electrode (in saturated KCl solution) as the reference electrode, a graphite rod as the counter electrode, and a GC electrode (5.61 mm in diameter for the RRDE test) modified by catalysts as the working electrode. To determine the reaction mechanism for OER, the ring potential was held constant at 1.5 V vs. RHE to oxidize HO₂⁻ intermediates in O₂-saturated 1 M KOH. To ensure that the

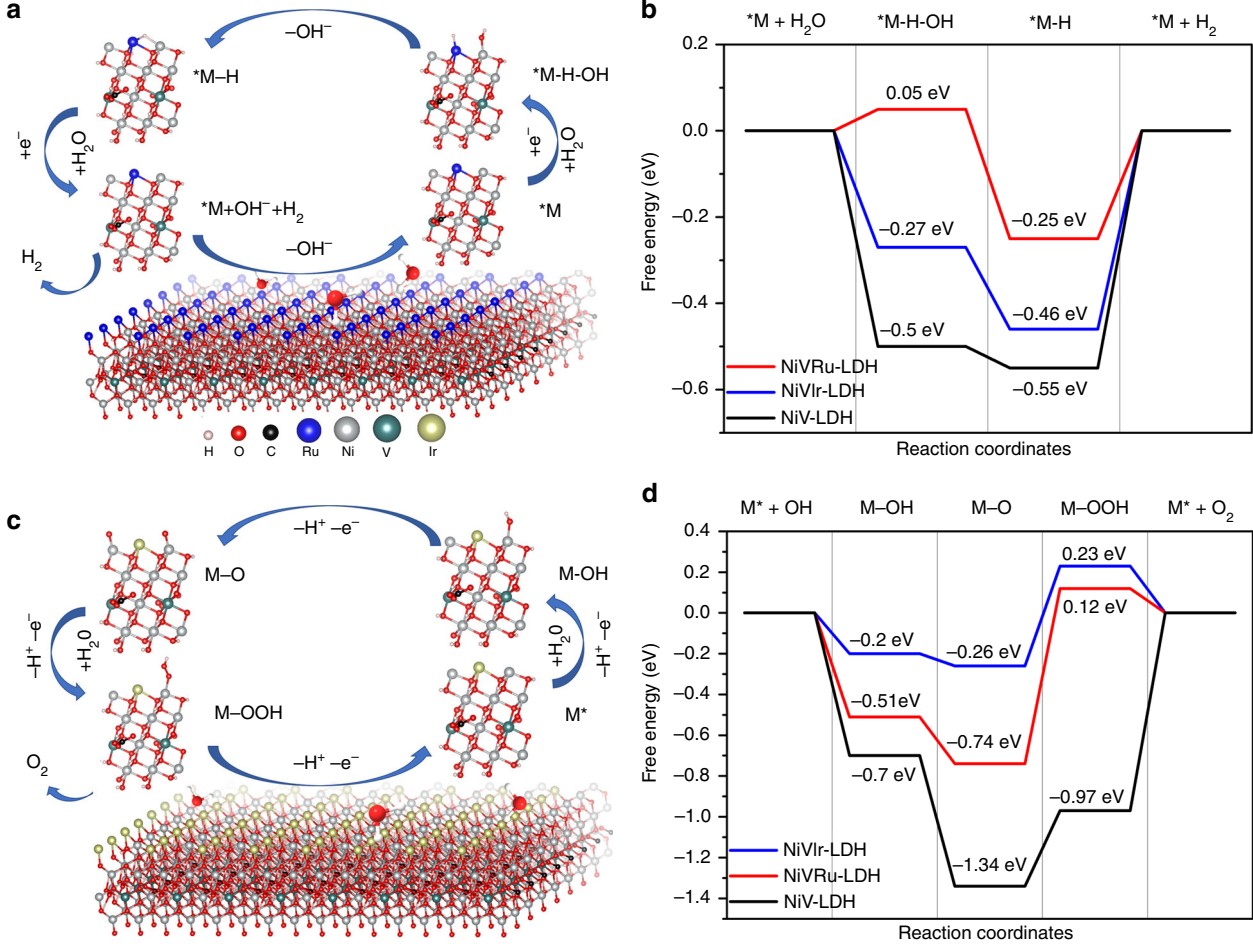

**Fig. 7** DFT calculations. **a** The atomic model of NiVRu-LDH and the proposed HER pathway. **b** The free energy diagram of HER on the NiV-LDH, NiVRu-LDH, and NiVIr-LDH catalysts. **c** The atomic model of NiVIr-LDH and the proposed OER pathway. **d** The free energy diagram of OER on the NiV-LDH, NiVRu-LDH, and NiVIr-LDH catalysts

oxidation current originated from oxygen evolution, the ring potential was held constant at 0.40 V vs. RHE to reduce the $O_2$ formed from the catalyst on the disk electrode in $N_2$-saturated 1 M KOH. The FE ($\varepsilon$) was calculated as follows:

$$\varepsilon = I_{ring}/(I_{disk} \times N) \qquad (1)$$

where $I_{ring}$ and $I_{disk}$ are the stabilized current densities obtained on the Pt ring and GC disc, respectively, and $N$ is the ring current collection efficiency, which was determined to be 0.2.

**Computational method.** All DFT calculations were performed by the Vienna Ab initio Simulation Package[68,69], employing the Projected Augmented Wave[70] method. The revised Perdew–Burke–Ernzerhof functional was used to describe the exchange and correlation effects[71–73]. The GGA + U calculations were performed using the model proposed by Dudarev et al.[74], with $U_{eff}$ ($U_{eff}$ = Coulomb $U$ − exchange $J$) values of 6.4, 3.5, 3, and 3.5 eV for Ni, V, Ru, and Ir, respectively. The layered hydroxides were constructed by using a $4 \times 2$ supercell, with one $CO_3$ layer separating two adjacent layers. For all geometry optimizations, the cutoff energy was set to 400 eV. Monkhorst–Pack grids ($2 \times 5 \times 4$)[75] were used to carry out the surface calculations in all the models.

In alkaline conditions, OER can occur via the following four elementary steps:

$$OH^- + * \rightarrow *OH + e^- \qquad (2)$$

$$*OH + OH^- \rightarrow *O + H_2O + e^- \qquad (3)$$

$$*O + OH^- \rightarrow *OOH + e^- \qquad (4)$$

$$*OOH + OH^- \rightarrow * + O_2 + H_2O + e^- \qquad (5)$$

where * denotes the active sites on the catalyst surface. Based on the above mechanism, the free energy of three intermediate states, *OH, *O, and *OOH, are

important for identifying a given material's OER activity. The computational hydrogen electrode model[76] was used to calculate the free energies of OER, based on which the free energy of an adsorbed species is defined as

$$\Delta G_{ads} = \Delta E_{ads} + \Delta E_{ZPE} - T\Delta S_{ads} \qquad (6)$$

where $\Delta E_{ads}$ is the electronic adsorption energy, $\Delta E_{ZPE}$ is the zero point energy difference between adsorbed and gaseous species, and $T\Delta S_{ads}$ is the corresponding entropy difference between these two states. The electronic binding energy is referenced to ½ $H_2$ for each H atom and ($H_2O-H_2$) for each O atom, plus the energy of the clean slab. The corrections for the zero point energy and entropy of the OER intermediates can be found in the Supplementary Information.

Similarly, for the HER, two elementary steps are involved in alkaline conditions:

$$H_2O + * \rightarrow *OH + *H \qquad (7)$$

$$*H + H_2O + e^- \rightarrow H_2 + * + OH^- \qquad (8)$$

The descriptor proposed by Norskov et al.[77] was used to describe the HER activity on a given catalyst surface; the corrections for the zero point energy and entropy of the HER intermediates can also be found in the supporting information.

## Data availability

The data that support the findings of this work are available from the corresponding author upon reasonable request. The source data underlying Figs 1a–c, 2c–f, 3a–f, 3g–h, 4a, d, 5a, d, 6a, c and Supplementary Figs 1, 2, 8, 10, 12, 14 are provided as a Source Data file.

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

## Acknowledgements

This work was supported by the National Natural Science Foundation of China (numbers 21435005 and 21627808) and Key Research Program of Frontier Sciences, Chinese Academy of Sciences (number QYZDY-SSW-SLH019).

## Author contributions

X.Y. led the project. D.W. designed and performed the majority of the experiments and obtained most of the results including material synthesis, characterization, and electrochemical tests. Q.L., C.H., and Q.Q.L. helped to polish the paper. D.W., Z.X., and X.Y. wrote the paper. All the authors have discussed the results and wrote the paper together.

## Additional information

**Competing interests:** The authors declare no competing interests.

