## [Peer Review File · Nature Communications]

Reviewers' Comments:

Reviewer #1:

Remarks to the Author:

LDHs offers numerous advantages such as: large surface-to-bulk ratios, controllable layered structure, tunable chemical composition with different cation ratios, hierarchical porosity facilitating the diffusion of water molecules and release of gaseous products, strong electrostatic interactions between layers and interlayer anions etc. Layered double hydroxide (LDHs) therefore shows promises for diverse applications that includes adsorption, photochemistry, supercapacitors, metal-air batteries, and drug delivery etc. Of particular interest, LDHs have recently attracted increased attention as an electrocatalyst for water-splitting applications. The outstanding problems with LDHs are to overcome the poor conductivity, low electron and charge transfer ability and insufficient active edge sites. The research articles that would invest to overcome these shortcomings are therefore of practical importance. The reviewer therefore enthusiastically and meticulously read the work presented in this article by Yang et al. As per author's claim, Ru and Ir doped NiV-LDH showed an impressive performance toward HER with low overpotential.

Impressed by the performance, the Reviewers was excited to see what new contribution this article is going to make in advancement of the science regarding LDH. In Reviewer opinions, the whole article reads more like merely reporting the data rather than explaining the science with the help of the data. In Reviewer's opinion, this work is a pretty much an incremental work of Sun et al (DOI: 10.1038/ncomms11981) in terms of materials synthesis, characterization and presentation of the data. This article therefore clearly lacks novelty up to a standard to be considered for being published in Nature Communication.

To improve the quality of this manuscript, here below are some of the specific comments that authors may consider.

1. Synthesis of NiV-LDH was prepared by following the same recipe that was proposed by Sun et al (DOI: 10.1038/ncomms11981). This is not a problem. However, authors claimed "for the first time" (see page 4, line 67). In synthesis of NiV, Sun et al used Urea (there was no explanation why urea was needed). The authors also copied the same, and obviously, there is no explanation why urea is required in synthesis of NiV. Crediting others people work is a good practice for the integrity of science.

2. Cooking NiV with additional ingredient such as Ru and Ir resulted in better performance. As an inquisitive scientist, the reviewer was more interested and expected to see what big changes were made by adding these Ru and Ir in morphology, electronic, and catalytic properties of NiV in atomic level. Although mentioned in the title of this article, there is no obvious evidences substantiated by experimental data about what kind of atomic and electronic modifications were brought by incorporating Ru and Ir in the NiV framework. A clear insight of the changed electronic structure and induced electronic properties would be of great scientific interest, and therefore, could buy the merit of novelty to be considered for this journal.

3. In page 5, line 95-96: Authors claimed NiVRu(Ir) are basically as same as NiV. However, a clear additional peak for NiVRu(Ir) can be observed between 70-80 degree in the XRD patterns (Fig. 1A). This indicates that there is a change in the crystal diffraction plane, perhaps due to changed atomic orientation. It is kind of expected as well. To specify the orientation of crystal plans, authors are suggested to carry out rocking XRD curves (pole figures) using GIWAXS.

A changed in atomic orientation is also observed in HRTEM images (Fig. G-I), although authors overlooked by indexing all of them as (0012). Reviewer is also confused what does this (0012) stands for, because a plane is represented by (hkl). Reviewer suggests to carry out SAED patterns.

4. Authors are suggested to use NEXAFS to probe the interfacial interaction by exploring the lateral structure of these multilayered nanosheets (see Nature Communications 5, 3783, 2014).

5. Authors are highly recommended to carry out Raman spectroscopy measurements to see the change (if any) in the vibrational spectra to confirm the phase integrity (see Energy & Environmental Science, 2018, 12, 572-581). The spatial distributions of different species in the compound structures are suggested to visualize by EELS mapping.

6. The morphology of NiVIr looks like agglomerated nanoparticles, while NiV and NiVRu showed

nanosheets like morphology (Fig. 1 D-I). Therefore, the BET surface area is supposed to be lowest in case of NiVIr. Authors are suggested to re-examine the morphology and BET measurements.

7. Measurements to count the electron transfer rate for OER could give more specific understanding how the incorporation of Ru/Ir may enhance the O₂ yields (see ACS Catalysis, 2016, 6, 4720-4728; Acc. Chem. Res. 2019, 52, 248-257).

8. Can author comments on the H₂:O₂ ratio produced by these LDHs? Authors are also suggested to comments on the TOF. Because TOF is more indicative for representing the stability and efficiency of an electrocatalyst.

9. Authors are suggested to extend the DFT calculation to show the interlayer electronic-coupling effect that would help to meaningfully explain the HER mechanism at atomic level (see Nature Communications 5, 3783, 2014).

10. The manuscript lacks of readability because of overall poor English. Before it qualifies for publication, authors need to improve the English carefully.

Reviewer #2:

Remarks to the Author:

The authors have introduced Ru or Ir to tailor the atomic and electronic structure of firstly synthesized self-supported NiV-LDH as the bifunctional electrocatalysts for improving the kinetics of water splitting. I think this process is very ingenious and interesting. Especially, this electrocatalyst exhibits outstanding HER and also good OER performance. In my opinion, the experimental design is rigorous and very interesting, and the manuscript was well written. Therefore, I highly recommend its acceptance after minor revision in this journal.

1. For the OER polarization curves, a scan rate of 5 mV/s is not so good since there is an obvious oxidation peak at around 1.40 V. It is better to use 1 mV/s and collect the curves scanning from high to low potentials.

2. Recently there are two excellent bifunctional electrocatalysts reported for overall water splitting: NiFe LDH @ defective graphene (Adv. Mater. 2017, 29, 1700017), FeP/Ni₂P (Nat. Commun. 2018, 9, 2551). Can the authors make a comparison among the catalytic activities of this catalyst in the manuscript and these two reported catalysts?

3. This electrocatalyst could also show excellent stability at large current densities for the HER and OER. Can the authors check it?

Reviewer #3:

Remarks to the Author:

In this manuscript, the authors reported a self-supported nickel-vanadium layered double hydroxide for water splitting. Further modulating atomic and electronic structure of NiV-LDH by Ru or Ir doping can significantly enhance the kinetics of HER or OER in alkaline media. For HER, the best performance was obtained to be Ru doping NiV-LDH (NiVRu-LDH) with Ru loading of 8.7 wt.%, resulting only 12 mV overpotential to deliver 10 mA cm⁻². On the other hand, NiVIr-LDH containing 12 wt% Ir, presents the optimum OER performance. First-principles calculation indicates that Ru doped NiVRu-LDH has lowest water dissociation energy barrier and hydrogen adsorption free energy, while Ir doping is more conducive to M-OH and M-O process. The authors have performed detailed structural characterizations and systemically electrochemical tests combined with the theoretical calculations. In my opinion, this paper is obviously innovative and significant to the field of water splitting and should be published in Nature Communication. In addition, minor revisions should be addressed before publication.

1. The authors referred that "the Ru (or Ir) replaces the V centers on NiV-LDH which..." (Line 71, page

- 4). Besides XPS results, could the author give the direct evidence for supporting this point in the text? For instance, the V vacancies appear and re-coordinate in the experimental process. Here is a recent paper should be helpful to refer to (Adv. Mater., 2019, 31, 1805581).
2. The location of diffraction peak corresponding to the (003) facet of NiV-LDH may not be at 11.2° in Fig. 1A, please double check it. Moreover, the authors claimed that the XRD patterns can be retained when Ru (or Ir) replaces the V centers in NiV-LDH, resulting in NiVRu-LDH (or NiVIr-LDH), but why there are strong Ni diffraction peaks in NiVRu-LDH and NiVIr-LDH?
3. In Fig. 4(a), why is the Pt/C tested in OER activity as the reference?
4. The detailed calculation process of active sites (n) for TOF should be provided.
5. The authors should re-consider the effects of defects induced by Ru- and Ir-doping. The defects on 2D materials are proved to be highly active for HER and OER. Some recent publications may be helpful for discussion, such as Chem, 2019, DOI:10.1016/j.chempr.2019.02.008; Adv. Mater., 2019, 31, 1805581; J. Am. Chem. Soc. 2018, 140, 34, 10757-10763.

Response to reviewers:

To reviewer #1:

Remarks to the Author:

LDHs offers numerous advantages such as: large surface-to-bulk ratios, controllable layered structure, tunable chemical composition with different cation ratios, hierarchical porosity facilitating the diffusion of water molecules and release of gaseous products, strong electrostatic interactions between layers and interlayer anions etc. Layered double hydroxide (LDHs) therefore shows promises for diverse applications that includes adsorption, photochemistry, supercapacitors, metal–air batteries, and drug delivery etc. Of particular interest, LDHs have recently attracted increased attention as an electrocatalyst for water-splitting applications. The outstanding problems with LDHs are to overcome the poor conductivity, low electron and charge transfer ability and insufficient active edge sites. The research articles that would invest to overcome these shortcomings are therefore of practical importance. The reviewer therefore enthusiastically and meticulously read the work presented in this article by Yang et al. As per author's claim, Ru and Ir doped NiV-LDH showed an impressive performance toward HER with low overpotential.

Impressed by the performance, the Reviewers was excited to see what new contribution this article is going to make in advancement of the science regarding LDH. In Reviewer opinions, the whole article reads more like merely reporting the data rather than explaining the science with the help of the data. In Reviewer's opinion, this work is a pretty much an incremental work of Sun et al (DOI: 10.1038/ncomms11981) in terms

of materials synthesis, characterization and presentation of the data. This article therefore clearly lacks novelty up to a standard to be considered for being published in Nature Communication.

To improve the quality of this manuscript, here below are some of the specific comments that authors may consider.

Reply: We are very grateful to the reviewers who read our article enthusiastically and meticulously, and ask these professional questions to help improve our work. We greatly admire the reviewers' rich professional knowledge and rigorous research attitude. We have carefully studied these comments and tried our best to solve every problem, if there are any deficiencies, please don't hesitate to let us know.

1. Synthesis of NiV-LDH was prepared by following the same recipe that was proposed by Sun et al (DOI: 10.1038/ncomms11981). This is not a problem. However, authors claimed “for the first time” (see page 4, line 67). In synthesis of NiV, Sun et al used Urea (there was no explanation why urea was needed). The authors also copied the same, and obviously, there is no explanation why urea is required in synthesis of NiV. Crediting others people work is a good practice for the integrity of science.

Reply: The synthesis method in Sun's article is just one of the many good work we have referenced in designing the urea hydrolysis method to synthesize NiV-LDH, not the only one. Different from the Sun's work, we said that we “for the first time” (we have modified our statement in the manuscript) synthesize the “self-supported” NiV-LDH, NiVRu-LDH and NiVIr-LDH on Ni foam, the morphology structures of our two work are quite different, and the self-supporting structure supported on Ni foam not only

prevents the catalyst from peeling off during long-term electrocatalysis but also provides more active sites.

The role of urea in the synthesis of LDH materials has long been explained by previous articles, so in recent years the article on the synthesis of LDH materials by urea hydrolysis method has rarely explained the role of urea in the text. Why urea is necessary in the synthesis of LDH materials can be explained as follow:

The chemical formula of LDH is $[M^{2+}_{1-x}M^{3+}_x(OH)_2]^{x+}(A^{n-})_{x/n} \cdot mH_2O$, where M^{2+} may be Mg, Zn, Co, Ni, Mn, etc.; M^{3+} may be Al, Cr, Fe, V and A^{n-} is the charged compensating anion (CO_3^{2-} , SO_4^{2-} , Cl^- , NO_3^- , organic anions, etc.). The common method for the synthesis of LDHs are prepared by co-precipitation of the chosen M^{2+} and M^{3+} hydroxides with diluted NaOH and/or $NaHCO_3$ or Na_2CO_3 solutions. The drawback of the method is that, no matter how fast the stirring speed is, the instantaneous pH value is certainly different in different parts of the slurry. As a result, it is very difficult to obtain LDH with high crystallinity. And this method often results in strong agglomeration of primary particles in aggregates with very broad size distributions. Such morphology leads to very low specific surface areas and porosity that may adversely affect the catalytic properties of LDH. Moreover, once the aggregates are formed they are very stable and resistant to de-cohesion even under powerful ultrasonic treatments. To predict the behavior of LDHs in the applications, the control and reproducibility of their crystal and particle properties is important and a high crystallinity is necessary. The urea hydrolysis method introduced by Costantino et al. was an important advancement in this regard. The urea method utilizes urea

instead of NaOH as the precipitating agent. The advantage of using urea is that the urea hydrolysis progresses slowly which leads to a low degree of super saturation during precipitation. Urea is a weak Bronsted base ($pK_b = 13.8$). It is highly soluble in water and its controlled hydrolysis in aqueous solutions can yield ammonium cyanate or its ionic form (NH_4^+ , NCO^-). Prolonged hydrolysis results in either CO_2 in an acidic medium or CO_3^{2-} in a basic environment as shown below:

A reaction temperature above 60 °C produces the progressive decomposition of urea in ammonium hydroxide leading to a homogeneous precipitation. This method has been already employed for the synthesis of well crystallized MAI-LDH (M = Li, Mg, Ni, Co), NiFe -LDH, CoTi -LDH and even three-component LDH with large particle sizes.

<Fig. R1 The SEM images of NiV-LDH described in the manuscript A, B), and the

samples without the addition of urea C, D) >

In order to clearly show the influence of urea on our work, we did a control experiment to synthesize NiV-LDH without the addition of urea. The morphology shown in Fig. R1C and D are quite different from that in Fig. R1A and B. The sample synthesized without urea look more like nanorod rather than nanosheet, so lack a good stereoscopic sense or layering sense.

Then we carry out the Fourier transform infrared (FTIR) spectroscopy analysis (Fig. R2), the stretching vibrations of intercalated carbonates in these samples are observed at ~ 680 and ~ 1380 cm^{-1} .

< Fig. R2 The FTIR spectra of the NiV-LDH, NiVRu-LDH and NiVIr-LDH. >

2. Cooking NiV with additional ingredient such as Ru and Ir resulted in better performance. As an inquisitive scientist, the reviewer was more interested and expected to see what big changes were made by adding these Ru and Ir in morphology, electronic,

and catalytic properties of NiV in atomic level. Although mentioned in the title of this article, there is no obvious evidences substantiated by experimental data about what kind of atomic and electronic modifications were brought by incorporating Ru and Ir in the NiV framework. A clear insight of the changed electronic structure and induced electronic properties would be of great scientific interest, and therefore, could buy the merit of novelty to be considered for this journal.

Reply: Inspired by the reviews' comments, in the past three month we re-examined the XRD patterns, re-measured the BET surface area, BJH pore size distribution and lattice spacing in the HRTEM images. And added the SAED patterns, Raman spectra, FTIR spectra, EELS Mapping, XANES and the Fourier and wavelet transform (FT/WT) analyses of the extended X-ray absorption fine structure (EXAFS). NiVRu-LDH and NiVIr-LDH feature an apparently smaller nanosheet size and charge transfer resistance than those of NiV-LDH. XRD and Raman spectra reveal possible lattice distortion caused by an isomorphic substitution of V by Ru (Ir), and V vacancies appear, which is confirmed by the EXAFS and WT analyses of EXAFS data, and the σ^2 results suggest a severely distorted octahedral V environment caused by Ru or Ir doping. XPS, EELS and XANES reveal the synergistic interaction among Ni, V and Ru (Ir) cations, derived from quite different valence electronic configurations.

We also detected the reaction mechanism of OER by a rotating ring-disk electrode (RRDE), and obtained the Faradaic efficiency (FE) of NiVRu-LDH and NiVIr-LDH for the HER and OER.

Please see the manuscript and this response for more detailed information.

3. In page 5, line 95-96: Authors claimed NiVRu(Ir) are basically as same as NiV. However, a clear additional peak for NiVRu(Ir) can be observed between 70-80 degree in the XRD patterns (Fig. 1A). This indicates that there is a change in the crystal diffraction plane, perhaps due to changed atomic orientation. It is kind of expected as well. To specify the orientation of crystal plans, authors are suggested to carry out rocking XRD curves (pole figures) using GIWAXS.

A changed in atomic orientation is also observed in HRTEM images (Fig. G-I), although authors overlooked by indexing all of them as (0012). Reviewer is also confused what does this (0012) stands for, because a plane is represented by (hkl). Reviewer suggests to carry out SAED patterns.

Reply: When we first used the NiV-LDH, NiVRu-LDH and NiVIr-LDH with Ni foam for XRD test, as shown in Fig. R3, the diffraction peak intensity of Ni foam is too strong to show the peak of materials. Then the NiV-LDH, NiVRu-LDH and NiVIr-LDH are scraped from the Ni foam for the XRD test, few Ni foam was inevitably mixed into the materials, so the Ni diffraction peaks at 44.3, 51.7 and 76.1° (a clear additional peak between 70-80) are attributed to the Ni foam. This phenomenon can be found in many articles (*Nat. Commun.*, 2018, 9, 2551, *Energy Environ. Sci.*, 2019, 12, 572.)

< Fig. R3 The XRD patterns of the NiV-LDH, NiVRu-LDH and NiVIr-LDH grown on the Ni foam. >

Thank you for the kind suggestions. We have performed the SAED patterns (Fig. R4), and no obvious change was observed which means the doping of Ru and Ir have no influence on the whole crystallinity because of the smaller content and similar atomic radius of V^{3+} (64 pm), Ni^{2+} (69 pm), Ir^{3+} (68 pm) and Ru^{3+} (68 pm). We also figure out the lattice spacing in HRTEM again which will index to the lattice plane of (018).

<Fig. R4 The HRTEM images and SAED patterns of A, D) NiV-LDH, B, E) NiVRu-LDH and C, F) NiVIr-LDH, respectively>

4. Authors are suggested to use NEXAFS to probe the interfacial interaction by exploring the lateral structure of these multilayered nanosheets (see Nature Communications 5, 3783, 2014).

Reply: We have meticulously read this article (see Nature Communications 5, 3783, 2014) published by the Qiao's group, and we have been studying and following the articles published by this group since a few years ago. For the first time, this article proposes a metal-free electrocatalyst for HER, the authors couple graphitic-carbon nitride with nitrogen-doped graphene to produce a metal-free hybrid catalyst, and the electrocatalytic HER performance of which is comparable or even better than that of traditional metallic catalysts.

By contrast, the metal-based electrocatalyst in our article contain many metallic and nonmetallic elements, combined with other articles about LDH materials, we carried out in-depth analyses on the NEXAFS.

To clarify the V vacancies and the incorporation effect of Ru (Ir) on the local atomic coordination and electronic structure of NiV-LDH, X-ray absorption near-edge structure (XANES) spectroscopy was conducted. As shown in Fig. R5A and B, the Ni K-edge spectra of the three samples are quite similar; however, there are significant differences in the V K-edge spectra, suggesting that the doping of Ru (Ir) has a stronger effect on V than that on Ni. Specifically, the V K-edge XANES spectra of the sample exhibit intense pre-edge peaks (inset in Fig. R5B), indicating the distorted coordination environment around V atoms in these materials. More interestingly, NiVRu-LDH and NiVIr-LDH show a higher pre-edge peak than that of NiV-LDH in the V K-edge XANES, implying that Ru (Ir) incorporation brings a higher degree of octahedral geometry distortion at the V sites in NiVRu-LDH and NiVIr-LDH compared to those in NiV-LDH. Similar results are found in the wavelet transform (WT) analyses of the Ni K-edge and V K-edge data (Fig. R5E and F). There are no obvious differences except for the weaker peak intensity at approximately 2.5-3.0 Å of NiVRu-LDH and NiVIr-LDH than that of NiV-LDH in the extended X-ray absorption fine structure (EXAFS) WT map, indicating the existence of Ni or V vacancies. In contrast, the peak intensities at approximately 3.0 Å of NiVRu-LDH and NiVIr-LDH are significantly less than that of NiV-LDH, which also shows that the change in V is larger than Ni. The different oscillation amplitudes in the corresponding Ni K-edge and V K-edge $k^3\chi(k)$ oscillation

curves (Fig. R6) reveal a structural change in the coordination environment of Ni and V atoms. Thus, the Ni K-edge and V K-edge *R*-space spectra (Fig. R5C and D, Table R1) provide detailed information about the coordination number (*C.N.*). The FT curves of the Ni *K*-edge data exhibit two prominent coordination peaks at 1.5 and 2.7 Å that are attributed to the Ni-O peak and Ni-Ni/V peak, and the *C.N.* values of Ni-Ni/V in NiVRu-LDH (4.6) and NiVIr-LDH (4.2) are slightly reduced compared to that in NiV-LDH (5.1). Similarly, the FT curves of the V *K*-edge data display prominent V-O peak at 1.3 Å in these three samples, and Ni-Ni/V peak at 2.97 Å in NiV-LDH, 2.93 Å in NiVRu-LDH and 2.84 Å in NiVIr-LDH. The *C.N.* values of V-Ni/V in NiVRu-LDH (3.8) and NiVIr-LDH (2.8) are obviously reduced compared to that in NiV-LDH (5.3); a larger σ^2 for the V-Ni/V of NiVRu-LDH (0.0172) and NiVIr-LDH (0.0186) was obtained compared to the corresponding value for NiV-LDH (0.0123), suggesting a severely distorted octahedral V-Ni/V environment. Together, these results confirm that V vacancies do exist in the material. The C K-edge XANES spectra in Fig. R5G show two peaks at approximately 285.4 and 292.7 eV, which can be assigned to $\pi^*C=C$ and σ^*C-C , respectively. Notably, NiVRu-LDH and NiVIr-LDH show a distinct decrease in peak intensity at approximately 288.6 eV (which is assigned to M-O-C bonds; M= Ni/V) compared to that of NiV-LDH. On the basis of both the XANES and EXAFS data for Ni and V above, the M here should be V. This clearly indicates the absence of V in NiVRu-LDH and NiVIr-LDH. Based on previous reports, another peak at approximately 290.2 eV is assigned to carbonate, which originates from the interlayer carbonate in NiV-LDH. This observation is consistent with the FTIR results. The

positions of the V L- and O K-edges make their spectra partially overlap, so the combined spectra of the V L- and O K-edges are presented in Fig. R5H. Three peaks (A, B and C) are observed in the O K-edge XANES spectra. Peak A at 530.5 eV and the broad peak C at 539 eV are assigned to $\pi^*C=O$ and σ^*C-O , respectively. Peak B at near 533.1 eV may be assigned to π^*C-O . In the V L-edge spectra, the two broad peaks centered at 519 and 525.9 eV are the L_3 and L_2 peaks, which are assigned to V $2p_{3/2}$ and V $2p_{1/2}$ transitions, respectively, and which are consistent with the XPS results. In sharp contrast, the peaks of L_2 and L_3 of V are noticeable in NiV-LDH but are almost absent in NiVRu-LDH and NiVIr-LDH, which strongly proves that vacancies of V in NiVRu-LDH and NiVIr-LDH exist.

< Fig. R5 (A) Ni and (B)V K-edge XANES spectra. (C) and (D) V K-edge extended XANES oscillation functions $k^3\chi(k)$. WT-EXAFS of (E) Ni and (F) V of the as-

prepared NiV-LDH, NiVRu-LDH and NiVIr-LDH (From left to right). (G) C K-edge XANES spectra. (H) V L-edge and O K-edge XANES spectra. >

< Fig. R6 Ni K-edge extended XANES oscillation functions $k^3\chi(k)$ of (A) NiV-LDH, (C) NiV-LDH and (E) NiV-LDH. V K-edge extended XANES oscillation functions

$k^3\chi(k)$ of (B) NiV-LDH, (D) NiV-LDH and (F) NiV-LDH. >

Table R1. Summary the fitting parameters of Ni and V *K*-edge EXFAS curves for the as-prepared NiV-LDH, NiVRu-LDH and NiVIr-LDH catalysts.

Sample	Path	C.N.	R (Å)	$\sigma^2 \times 10^3$ (Å ²)	ΔE (eV)	R factor
NiV-LDH	Ni-O	6.4±0.6	2.04±0.01	8.2±0.8	-6.4±1.1	0.005
	Ni-Ni/V	5.1±0.7	3.09±0.01	9.6±1.0	0.4±1.3	
NiVRu-LDH	Ni-O	6.5±0.5	2.03±0.01	8.4±0.8	-6.6±1.0	0.004
	Ni-Ni/V	4.6±0.6	3.08±0.01	9.8±1.0	0.8±1.3	
NiVIr-LDH	Ni-O	6.5±0.6	2.03±0.01	8.2±0.9	-6.1±1.2	0.006
	Ni-Ni/V	4.2±0.8	3.08±0.01	9.5±1.3	0.8±1.7	
NiV-LDH	V-O	5.7±0.6	1.68±0.01	6.7±0.9	1.8±1.6	0.008
	V-Ni/V	5.3±2.5	3.40±0.03	12.3±3.8	5.6±0.6	
NiVRu-LDH	V-O	6.5±0.6	1.68±0.01	7.1±0.9	2.5±1.5	0.008
	V-Ni/V	3.8±2.3	3.37±0.04	17.2±5.6	3.7±4.8	
NiVIr-LDH	V-O	6.4±0.9	1.66±0.01	8.7±1.3	-2.2±2.3	0.014
	V-Ni/V	2.8±2.5	3.37±0.05	18.6±9.4	2.5±6.8	

Note: ΔE , inner potential correction; σ^2 , Debye Waller factor to account for both thermal and structural disorders; *R*-factor, indicating the goodness of the fit.

5. Authors are highly recommended to carry out Raman spectroscopy measurements to see the change (if any) in the vibrational spectra to confirm the phase integrity (see Energy & Environmental Science, 2018, 12, 572-581). The spatial distributions of

different species in the compound structures are suggested to visualize by EELS mapping.

Reply: Thank you for the kind suggestions. The Raman spectroscopy is performed to obtain information about the chemical identities of these samples as in Fig. R7A. The NiV-LDH before and after the Ru (Ir) doping exhibit the similar spectrum with mainly the strong peak $\sim 810\text{ cm}^{-1}$, due to the V-O vibration, which consistent with the results of FTIR (the main peak is also located at $\sim 800\text{ cm}^{-1}$). The Raman responses of V-O vibration for NiVRu-LDH and NiVIr-LDH are obviously weaker to that of NiV-LDH because of the replacement of V by Ru (Ir), besides, some of the peaks in Fig. R7B show a certain degree of red shift, and some peaks disappeared, which can be attributed to two reasons: 1. Size effects. The Raman peaks become weaker and blunter and shift slightly to lower wavenumbers as the grain size decreases, so NiVRu-LDH and NiVIr-LDH may have a smaller particle size than that of NiV-LDH. 2. Doping effect and defects. The defects could be edges, dislocations, cracks or vacancies in the sample. We suspect that this is due to the V vacancies which consistent with the XRD results and will be further verified later.

< Fig. R7 The Raman spectra of NiV-LDH, NiVRu-LDH and NiVIr-LDH >

The spatial distribution of every element in NiVRu-LDH and NiVIr-LDH is further identified by electron energy loss spectroscopy (EELS) and Fig. R8 indicate the homogenous Ru and Ir doping in the nanosheets.

< Fig. R8 HAADF-STEM image of the (A) NiVRu-LDH and (B) NiVIr-LDH and corresponding EELS mappings images.>

6. The morphology of NiVIr looks like agglomerated nanoparticles, while NiV and NiVRu showed nanosheets like morphology (Fig. 1 D-I). Therefore, the BET surface area is supposed to be lowest in case of NiVIr. Authors are suggested to re-examine the morphology and BET measurements.

Reply: As suggested, the morphology and specific surface area of NiV-LDH, NiVRu-LDH and NiVIr-LDH have been re-examined for several times. The morphology of the three samples is the same as that described in manuscript, however, the specific surface area is different from that we measured for the first time, but it is the same as the reviewer speculated. The results of the BET measurement are show as below. The specific surface area of NiV-LDH, NiVRu-LDH and NiVIr-LDH is 36.4, 32.6 and 15.4

$\text{m}^2 \text{g}^{-1}$, respectively (R9), the BET surface area of NiVIr-LDH is indeed the smallest in all three samples. Considering this result is different from that of the first time, we have patiently measured and confirmed it for many times and got the similar data. To further verify the authenticity of this result, we synthesized the NiV-LDH, NiVRu-LDH and NiVIr-LDH without using the Ni foam, and collect the powder sample to test their specific surface area, the result is shown in Fig. R10. The specific surface area of NiV-LDH (without Ni foam) and NiVRu-LDH (without Ni foam) is 324.1 and $264.4 \text{ m}^2 \text{g}^{-1}$, respectively, while the NiVIr-LDH (without Ni foam) also exhibits the smallest BET surface area of $189.1 \text{ m}^2 \text{g}^{-1}$.

< Fig. R9 Nitrogen adsorption-desorption isotherm and the corresponding pore size distribution of (A, B) NiV-LDH, (C, D) NiVRu-LDH and (E, F) NiVIr-LDH, respectively >

< Fig. R10 Nitrogen adsorption-desorption isotherm and the corresponding pore size distribution of (A, B) NiV-LDH, (C, D) NiVRu-LDH and (E, F) NiVIr-LDH without Ni foam, respectively >

7. Measurements to count the electron transfer rate for OER could give more specific understanding how the incorporation of Ru/Ir may enhance the O₂ yields (see ACS Catalysis, 2016, 6, 4720-4728; Acc. Chem. Res. 2019, 52, 248-257).

Reply: After carefully studied the two articles recommended by reviewers and combined with some other articles published by Qiao's groups, we used the rotating ring-disk electrode (RRDE) to get more detailed information about the reaction mechanism during the OER process. The RRDE with a Pt ring electrode potential of 1.50 V versus RHE was used to detect whether the peroxide intermediates formed during the OER process. As shown in Fig. R11A below, no detectable current from peroxide oxidation at the Pt ring was observed, suggesting negligible formation of peroxide intermediates and therefore a desirable four-electron pathway for water oxidation. The RRDE can also be used for quickly screening the approximate Faradaic efficiency (ϵ) of O_2 production (ϵ) (Fig. R11B). We employed the RRDE in N_2 -saturated 1 M KOH with a ring potential of 0.40 V to reduce the O_2 molecular, rendering a continuous OER (disk electrode) \rightarrow ORR (oxygen reduction reaction, ring electrode) process. With the disk current of 700 μA , O_2 molecules generated from the NiVIr-LDH surface on the disk electrode sweep across the surrounding Pt ring electrode held at an ORR potential, and are rapidly reduced. Consequently, a ring current of approximately 140 μA ($700 \mu A \times 0.2$; RRDE collecting efficiency $N=0.2$) was detected, suggesting the oxidation current can be fully attributed to the OER with a high ϵ of $> 99\%$.

< Fig. R11 A The ring current of NiVIr-LDH on a RRDE (1500 rpm) in O₂-saturated 1 M KOH solution (ring potential: 1.50 V). Fig. B The ring current of NiVIr-LDH on a RRDE (1500 rpm) in N₂-saturated 1 M KOH solution (ring potential: 0.40 V). >

8. Can author comments on the H₂:O₂ ratio produced by these LDHs? Authors are also suggested to comments on the TOF. Because TOF is more indicative for representing the stability and efficiency of an electrocatalyst.

Reply: The Faradaic efficiency (FE) of NiVRu-LDH and NiVIr-LDH for the HER and OER was measured quantitatively from the total amount of charge passed through the cell during electrolysis and the total amount of evolved gas recorded by the pressure sensor (Fig. R12). The amount of experimentally generated H₂ and O₂ matches well with the theoretically calculated amount under the total charge during the electrolysis process, suggesting that the FE is close to 100 % for the HER and OER, with the ratio of H₂ and O₂ being close to 2:1.

The TOF was also commented on the manuscript.

< Fig. R12 The amount of gas theoretically calculated and experimentally measured versus time for HER and OER of NiVRu-LDH and NiVIr-LDH, respectively. >

9. Authors are suggested to extend the DFT calculation to show the interlayer electronic-coupling effect that would help to meaningfully explain the HER mechanism at atomic level (see Nature Communications 5, 3783, 2014).

Reply: As described in the article (Nature Communications 5, 3783, 2014), after coupling g-C₃N₄ with N-graphene, the charge density in hybrid's interlayer was redistributed in the form of an apparent electron transfer from conductive N-graphene to g-C₃N₄, leading to an electron-rich region on g-C₃N₄ layer and a hole-rich region on N-graphene layer, the interlayer electronic-coupling effect enhanced electron mobility in the C₃N₄@NG, which is significant for the electrocatalytic HER.

The pure LDHs materials are two-dimensional anionic clays consisting of positively charged brucite-like host layers and exchangeable charge-balancing interlayer anions,

the interlayer anion in our materials is CO_3^{2-} , so there is no interlayer electronic-coupling effect between the brucite-like host layers in the NiV-LDH. However, the interlayer electronic-coupling effect could be found in the 3D hierarchical composite nanostructures of LDHs, which included 3D hierarchical structures LDH/Carbon-based materials, 3D hierarchical nanostructures of LDHs/Metal and 3D hierarchical nanostructures LDHs/Metal compounds. For example, NiFe-LDH/Co, N-CNF (Adv. Energy Mater. 2017, 7, 1700467), $\text{Co}_{0.85}\text{Se}/\text{NiFe}$ LDH/graphene (Energy Environ. Sci. 2016, 9, 478) and NiFe LDH@NiCoP/NF (Adv. Funct. Mater. 2018, 28, 1706847).

10. The manuscript lacks of readability because of overall poor English. Before it qualifies for publication, authors need to improve the English carefully.

Reply: We have revised the grammar, sentence patterns, punctuation of the article many times, and asked the English language editing service from American Journal Experts (AJE) for help, we hope it qualifies for publication.

To reviewer #2:

Remarks to the Author:

The authors have introduced Ru or Ir to tailor the atomic and electronic structure of firstly synthesized self-supported NiV-LDH as the bifunctional electrocatalysts for improving the kinetics of water splitting. I think this process is very ingenious and interesting. Especially, this electrocatalyst exhibits outstanding HER and also good OER performance. In my opinion, the experimental design is rigorous and very interesting, and the manuscript was well written. Therefore, I highly recommend its acceptance after minor revision in this journal.

Reply: Thanks for your positive comments on our work and supporting its publication.

We have modified our work based on your constructive comments.

1. For the OER polarization curves, a scan rate of 5 mV/s is not so good since there is an obvious oxidation peak at around 1.40 V. It is better to use 1 mV/s and collect the curves scanning from high to low potentials.

Reply: As suggested, the OER polarization curves are collected scanning from high to low potentials with scan rate of 1 mV/s (Figure R13).

< Fig. R13 OER linear sweeping voltammetry curves of different electrocatalysts in 1 M KOH solution. >

2. Recently there are two excellent bifunctional electrocatalysts reported for overall water splitting: NiFe LDH @ defective graphene (Adv. Mater. 2017, 29, 1700017), FeP/Ni₂P (Nat. Commun. 2018, 9, 2551). Can the authors make a comparison among the catalytic activities of this catalyst in the manuscript and these two reported catalysts?

Reply: NiFe LDH@defective graphene and FeP/Ni₂P are two state-of-the-art bifunctional catalysts for overall water splitting, what commendable is that they are all non-noble metal catalysts. The comparison among the catalytic activities of the two catalysts and our catalysts are performed in the manuscript.

3. This electrocatalyst could also show excellent stability at large current densities for the HER and OER. Can the authors check it?

Reply: Indeed, NiVRu-LDH and NiVIr-LDH can also show excellent stability at large current densities for the HER and OER, respectively. For NiVRu-LDH, a long-term electrocatalytic HER process was carried out at current densities of 50 and 200 mA cm⁻², the steady HER overpotential was retained for 200 h (Figure R14A). The NiVIr-LDH displays long-term electrochemical durability for 600 h at 50 mA cm⁻² and 400 h at 200

mA cm^{-2} , respectively (Figure R14B).

< Fig. R14 A The chronopotentiometric curves of NiVRu-LDH for the HER test at 50 and 200 mA cm^{-2} . B The chronopotentiometric curves of NiVIr-LDH for the OER test at 50 and 200 mA cm^{-2} .>

To reviewer #3:

Remarks to the Author:

In this manuscript, the authors reported a self-supported nickel-vanadium layered double hydroxide for water splitting. Further modulating atomic and electronic structure of NiV-LDH by Ru or Ir doping can significantly enhance the kinetics of HER or OER in alkaline media. For HER, the best performance was obtained to be Ru doping NiV-LDH (NiVRu-LDH) with Ru loading of 8.7 wt.%, resulting only 12 mV

overpotential to deliver 10 mA cm⁻². On the other hand, NiV_{Ir}-LDH containing 12 wt% Ir, presents the optimum OER performance. First-principles calculation indicates that Ru doped NiVRu-LDH has lowest water dissociation energy barrier and hydrogen adsorption free energy, while Ir doping is more conducive to M-OH and M-O process. The authors have performed detailed structural characterizations and systemically electrochemical tests combined with the theoretical calculations. In my opinion, this paper is obviously innovative and significant to the field of water splitting and should be published in Nature Communication. In addition, minor revisions should be addressed before publication.

Reply: Thank you very much for your positive comments and support of our work.

1. The authors referred that “the Ru (or Ir) replaces the V centers on NiV-LDH which...” (Line 71, page 4). Besides XPS results, could the author give the direct evidence for supporting this point in the text? For instance, the V vacancies appear and re-coordinate in the experimental process. Here is a recent paper should be helpful to refer to (Adv. Mater., 2019, 31, 1805581).

Reply: Thank you very much for giving us a new idea to explore the effects of the addition of Ru and Ir on the atomic and electronic structure of materials. The article (Adv. Mater., 2019, 31, 1805581) was well written and it gives us a lot of inspiration. Besides XPS results, XRD pattern and Raman spectra reveal possible lattice distortion caused by an isomorphic substitution of V by Ru (Ir), and V vacancies appear, which is confirmed by the EXAFS and WT analyses of EXAFS data, and the σ^2 results suggest

a severely distorted octahedral V environment caused by Ru or Ir doping. The results of XPS, EELS and XANES reveal the synergistic interaction among Ni, V and Ru (Ir) cations, derived from quite different valence electronic configurations.

XRD: The X-ray diffraction (XRD) patterns of NiV-LDH, NiVRu-LDH and NiVIr-LDH are shown in Fig. R15. The diffraction peaks at 11.6, 22.8, 33.5 and 60.5° can be indexed to the characteristic (003), (006), (009) and (110) facets of NiV-LDH²³⁻²⁴. With the incorporation of Ru (Ir), no additional diffraction peaks emerge that are associated with the formation of other phases. However, the increased diffraction peak width and reduced peak intensity reveal imperfections in the layers, lattice distortion may be caused by the isomorphic substitution of V by Ru (Ir), and V vacancies appear and re-coordinate in the experimental process.

< Fig. R15 The XRD patterns of NiV-LDH, NiVRu-LDH and NiVIr-LDH scraped from Ni foam. >

Raman: Additionally, Raman spectroscopy was performed to obtain information about the chemical identities of these samples (Fig. R16). The NiV-LDH samples before and after Ru (Ir) doping exhibit similar spectra, with a strong main peak at $\sim 810 \text{ cm}^{-1}$ due to V-O vibration, which is consistent with the FTIR results (the main peak is also located at $\sim 800 \text{ cm}^{-1}$). The Raman responses of V-O vibrations for NiVRu-LDH and NiVIr-LDH are clearly weaker than that of NiV-LDH because of the replacement of V by Ru (Ir). Additionally, some of the peaks in inset of Fig. R16B show a certain degree of redshift, and some peaks disappear, which can be attributed to two reasons: 1. Size effects. The Raman peaks become weaker and blunter and shift slightly to lower wavenumbers as the grain size decreases, so NiVRu-LDH and NiVIr-LDH may have a smaller particle size than that of NiV-LDH. 2. Doping effect and defects. Defects could be edges, dislocations, cracks or vacancies in a sample. We suspect that the presence of defects is due to V vacancies, which is consistent with the XRD results and will be further verified later.

< Fig. R16 The Raman spectra of NiV-LDH, NiVRu-LDH and NiVIr-LDH >

XANES: To clarify the V vacancies and the incorporation effect of Ru (Ir) on the local

atomic coordination and electronic structure of NiV-LDH, X-ray absorption near-edge structure (XANES) spectroscopy was conducted. As shown in Fig. R17A and B, the Ni K-edge spectra of the three samples are quite similar; however, there are significant differences in the V K-edge spectra, suggesting that the doping of Ru (Ir) has a stronger effect on V than that on Ni. Specifically, the V K-edge XANES spectra of the sample exhibit intense pre-edge peaks (inset in Fig. R17B), indicating the distorted coordination environment around V atoms in these materials. More interestingly, NiVRu-LDH and NiVIr-LDH show a higher pre-edge peak than that of NiV-LDH in the V K-edge XANES, implying that Ru (Ir) incorporation brings a higher degree of octahedral geometry distortion at the V sites in NiVRu-LDH and NiVIr-LDH compared to those in NiV-LDH. Similar results are found in the wavelet transform (WT) analyses of the Ni K-edge and V K-edge data (Fig. R17E and F). There are no obvious differences except for the weaker peak intensity at approximately 2.5-3.0 Å of NiVRu-LDH and NiVIr-LDH than that of NiV-LDH in the extended X-ray absorption fine structure (EXAFS) WT map, indicating the existence of Ni or V vacancies. In contrast, the peak intensities at approximately 3.0 Å of NiVRu-LDH and NiVIr-LDH are significantly less than that of NiV-LDH, which also shows that the change in V is larger than Ni. The different oscillation amplitudes in the corresponding Ni K-edge and V K-edge $k^3\chi(k)$ oscillation curves (Fig. R18) reveal a structural change in the coordination environment of Ni and V atoms. Thus, the Ni K-edge and V K-edge *R*-space spectra (Fig. R17C and D, Table 2) provide detailed information about the coordination number (*C.N.*). The FT curves of the Ni K-edge data exhibit two prominent coordination peaks

at 1.5 and 2.7 Å that are attributed to the Ni-O peak and Ni-Ni/V peak, and the *C.N.* values of Ni-Ni/V in NiVRu-LDH (4.6) and NiVIr-LDH (4.2) are slightly reduced compared to that in NiV-LDH (5.1). Similarly, the FT curves of the V *K*-edge data display prominent V-O peak at 1.3 Å in these three samples, and Ni-Ni/V peak at 2.97 Å in NiV-LDH, 2.93 Å in NiVRu-LDH and 2.84 Å in NiVIr-LDH. The *C.N.* values of V-Ni/V in NiVRu-LDH (3.8) and NiVIr-LDH (2.8) are obviously reduced compared to that in NiV-LDH (5.3); a larger σ^2 for the V-Ni/V of NiVRu-LDH (0.0172) and NiVIr-LDH (0.0186) was obtained compared to the corresponding value for NiV-LDH (0.0123), suggesting a severely distorted octahedral V-Ni/V environment. Together, these results confirm that V vacancies do exist in the material. The C *K*-edge XANES spectra in Fig. R17G show two peaks at approximately 285.4 and 292.7 eV, which can be assigned to $\pi^*C=C$ and σ^*C-C , respectively. Notably, NiVRu-LDH and NiVIr-LDH show a distinct decrease in peak intensity at approximately 288.6 eV (which is assigned to M-O-C bonds; M= Ni/V) compared to that of NiV-LDH. On the basis of both the XANES and EXAFS data for Ni and V above, the M here should be V. This clearly indicates the absence of V in NiVRu-LDH and NiVIr-LDH. Based on previous reports, another peak at approximately 290.2 eV is assigned to carbonate, which originates from the interlayer carbonate in NiV-LDH. This observation is consistent with the FTIR results. The positions of the V L- and O *K*-edges make their spectra partially overlap, so the combined spectra of the V L- and O *K*-edges are presented in Fig. R17H. Three peaks (A, B and C) are observed in the O *K*-edge XANES spectra. Peak A at 530.5 eV and the broad peak C at 539 eV are assigned to $\pi^*C=O$ and σ^*C-O , respectively. Peak

B at near 533.1 eV may be assigned to π^*C-O . In the V L-edge spectra, the two broad peaks centered at 519 and 525.9 eV are the L_3 and L_2 peaks, which are assigned to V $2p_{3/2}$ and V $2p_{1/2}$ transitions, respectively, and which are consistent with the XPS results. In sharp contrast, the peaks of L_2 and L_3 of V are noticeable in NiV-LDH but they are almost absent in NiVRu-LDH and NiVIr-LDH, which strongly proves that vacancies of V in NiVRu-LDH and NiVIr-LDH exist.

< Fig. R17 (A) Ni and (B) V K-edge XANES spectra. (C) Ni and (D) V K-edge extended XANES oscillation functions $k^3\chi(k)$. WT-EXAFS of (E) Ni and (F) V of the as-prepared NiV-LDH, NiVRu-LDH and NiVIr-LDH (From left to right). (G) C K-edge XANES spectra. (H) V L-edge and O K-edge XANES spectra. >

< Fig. R18 Ni K-edge extended XANES oscillation functions $k^3\chi(k)$ of (A) NiV-LDH, (C) NiV-LDH and (E) NiV-LDH. V K-edge extended XANES oscillation functions $k^3\chi(k)$ of (B) NiV-LDH, (D) NiV-LDH and (F) NiV-LDH. >

Table 2. Summary the fitting parameters of Ni and V *K*-edge EXFAS curves for the as-prepared NiV-LDH, NiVRu-LDH and NiVIr-LDH catalysts.

Sample	Path	C.N.	R (Å)	$\sigma^2 \times 10^3$ (Å ²)	ΔE (eV)	R factor
NiV-LDH	Ni-O	6.4±0.6	2.04±0.01	8.2±0.8	-6.4±1.1	0.005
	Ni-Ni/V	5.1±0.7	3.09±0.01	9.6±1.0	0.4±1.3	
NiVRu-LDH	Ni-O	6.5±0.5	2.03±0.01	8.4±0.8	-6.6±1.0	0.004
	Ni-Ni/V	4.6±0.6	3.08±0.01	9.8±1.0	0.8±1.3	
NiVIr-LDH	Ni-O	6.5±0.6	2.03±0.01	8.2±0.9	-6.1±1.2	0.006
	Ni-Ni/V	4.2±0.8	3.08±0.01	9.5±1.3	0.8±1.7	
NiV-LDH	V-O	5.7±0.6	1.68±0.01	6.7±0.9	1.8±1.6	0.008
	V-Ni/V	5.3±2.5	3.40±0.03	12.3±3.8	5.6±0.6	
NiVRu-LDH	V-O	6.5±0.6	1.68±0.01	7.1±0.9	2.5±1.5	0.008
	V-Ni/V	3.8±2.3	3.37±0.04	17.2±5.6	3.7±4.8	
NiVIr-LDH	V-O	6.4±0.9	1.66±0.01	8.7±1.3	-2.2±2.3	0.014
	V-Ni/V	2.8±2.5	3.37±0.05	18.6±9.4	2.5±6.8	

Note: ΔE , inner potential correction; σ^2 , Debye Waller factor to account for both thermal and structural disorders; *R*-factor, indicating the goodness of the fit.

2. The location of diffraction peak corresponding to the (003) facet of NiV-LDH may not be at 11.2 ° in Fig. 1A, please double check it. Moreover, the authors claimed that the XRD patterns can be retained when Ru (or Ir) replaces the V centers in NiV-LDH, resulting in NiVRu-LDH (or NiVIr-LDH), but why there are strong Ni diffraction peaks in NiVRu-LDH and NiVIr-LDH?

Reply: The diffraction peak corresponding to the (003) facet of NiV-LDH in Fig. 1A is at 11.6 °, we have corrected it in the manuscript, thanks for your meticulously reading. When we first used the NiV-LDH, NiVRu-LDH and NiVIr-LDH with Ni foam for XRD test, as shown in Fig. R19, the diffraction peak intensity of Ni foam is too strong to show the peak of materials. Then the NiV-LDH, NiVRu-LDH and NiVIr-LDH are

scraped from the Ni foam for the XRD test, few Ni foam was inevitably mixed into the materials, so the Ni diffraction peaks at 44.3, 51.7 and 76.1° are attributed to the Ni foam. This phenomenon can be found in many articles (*Nat. Commun.*, 2018, 9, 2551, *Energy Environ. Sci.*, 2019, 12, 572.)

<Fig. R19 The XRD patterns of the NiV-LDH, NiVRu-LDH and NiVIr-LDH grown on the Ni foam. >

3. In Fig. 4(a), why is the Pt/C tested in OER activity as the reference?

Reply: We are sorry to make this mistake, RuO₂ in Fig. R20 was wrongly written as

Pt/C, we have corrected it as shown in the following picture.

<Fig. R20 OER polarization curves of the as obtained NiV-LDH, NiVRu-LDH, NiVIr-LDH, Ni foam and RuO₂. >

4. The detailed calculation process of active sites (n) for TOF should be provided.

Reply: The method we used to calculate n and TOF is a common method which can be seen in many articles (*Nat. Commun.*, 2018, 9, 2551. *Nat. Nanotech.*, 2017, 12, 441.).

<Fig. R21 CV curves for NiVRu-LDH recorded between -0.2 V and 0.6 V vs. RHE in 1.0 M PBS (pH=7) at a scan rate of 0.05 mV s⁻¹. >

Since the difficulty in attributing the observed peaks to a given redox couple, the number of active sites should be proportional to the integrated charge over the CV curve. Assuming a one-electron process for both reduction and oxidation, the upper limit of active sites (n) for NiVRu-LDH could be calculated according to the follow equation:

$$n = Q/2F$$

where F=96485.3 C/mol and Q are the Faraday constant and the whole charge of CV curve, respectively (Figure R21). By this equation and the CV curves, taking NiVRu-LDH as an example, the detailed calculation process of n can be provided as follows:

$$Q = \frac{\int vA}{v} = \frac{0.00544}{0.05} = 0.1088 \text{ C}$$

$$n = \frac{Q}{2F} = \frac{0.1088}{2 \times 96485.3} = 5.64 \times 10^{-7} \text{ mol}$$

5. The authors should re-consider the effects of defects induced by Ru- and Ir-doping.

The defects on 2D materials are proved to be highly active for HER and OER. Some recent publications may be helpful for discussion, such as Chem, 2019, DOI:10.1016/j.chempr.2019.02.008; Adv. Mater., 2019, 31, 1805581; J. Am. Chem. Soc. 2018, 140, 34, 10757-10763.

Reply: By learning these references, the effects of defects induced by Ru- and Ir-doping on 2D NiV-LDH have been discussed in the manuscript.

REVIEWERS' COMMENTS:

Reviewer #1 (Remarks to the Author):

The Reviewer is happy to see that the authors have carried out the suggested characterizations, and addressed the comments satisfactorily. This revised version of the manuscript is much better than the original version. I am therefore in favor of its publication in Nature Communications after the following minor corrections:

1. I suggest to add the 'role of urea in LDH synthesis' as a supplementary note.
2. To be consistent, please number the supplementary tables as Table X or Table SX (where X = 1, 2, 3...), and revise the manuscript accordingly.

Reviewer #2 (Remarks to the Author):

The authors have addressed all of my concerns. I would like to suggest its acceptance in this journal.

Reviewer #3 (Remarks to the Author):

The authors addressed all my concerns satisfactorily. I recommend it published in Nat Comm. as its current version.

Response to reviewers:

To reviewer #1:

Remarks to the Author:

The Reviewer is happy to see that the authors have carried out the suggested characterizations, and addressed the comments satisfactorily. This revised version of the manuscript is much better than the original version. I am therefore in favor of its publication in Nature Communications after the following minor corrections:

Reply: The quality of our articles has been greatly improved with the help of reviewers, and we have learned a lot of expertise and skills in this revision process, thanks again for the reviewers.

1. I suggest to add the 'role of urea in LDH synthesis' as a supplementary note.

Reply: The 'role of urea in LDH synthesis' has been added in the supporting information.

2. To be consistent, please number the supplementary tables as Table X or Table SX (where X = 1, 2, 3...), and revise the manuscript accordingly.

Reply: We have corrected it in the revised manuscript.

To reviewer #2:

Remarks to the Author:

The authors have addressed all of my concerns. I would like to suggest its acceptance in this journal.

Reply: Thank you very much for agreeing the publication of our work.

To reviewer #3:

Remarks to the Author:

The authors addressed all my concerns satisfactorily. I recommend it published in Nat Comm. as its current version.

Reply: Thank you very much for agreeing the publication of our work.